# CONTRACTIVE ERROR FEEDBACK FOR GRADIENT COMPRESSION

## ABSTRACT

On-device memory concerns in distributed deep learning are becoming more severe due to i) the growth of model size in multi-GPU training, and ii) the adoption of neural networks for federated learning on IoT devices with limited storage. In such settings, this work deals with memory issues emerging with communication efficient methods. To tackle associated challenges, key advances are that i) instead of EFSGD that inefficiently manages memory, the sweet spot of convergence and memory usage can be attained via what is here termed contractive error feedback (ConEF); and, ii) communication efficiency in ConEF should be achieved by biased and allreducable gradient compression. ConEF is validated on various learning tasks that include image classification, language modeling, and machine translation. ConEF saves 80% – 90% of the extra memory in EFSGD with almost no loss on test performance, while also achieving 1.3x – 5x speedup of SGD.

## 1 INTRODUCTION

Many modern machine learning tasks can be cast as an optimization problem

$$\min_{\mathbf{x} \in \mathbb{R}^d} f(\mathbf{x}) := \frac{1}{N} \sum_{i=1}^{N} \mathbb{E}_{\xi \sim \mathcal{D}} \big[ f_i(\mathbf{x}, \xi) \big]. \tag{1}$$

Typically, the possibly nonconvex loss function $f_i(\mathbf{x}, \xi)$ is continuously differentiable with respect to (w.r.t.) $\mathbf{x}$. We assume that $f^* > -\infty$ is a global minimum of (1). In this work, we will focus on training neural networks (NN) by solving (1) in two important scenarios: i) data parallelism in a multi-GPU setting, and ii) federated learning where clients could be edge devices such as smart phones. In both cases, $N$ denotes the number of workers (GPUs or devices) and $f^*$ is no smaller than 0 given common choices of loss functions such as cross entropy and mean square.

While training NNs is a resource-hungry task, *bandwidth* and *memory* prevent us from fully harvesting the merits of parallelism. Limited bandwidth slows down the speed of information (e.g., gradients) exchange among workers. In multi-GPU settings, it is observed that communication of gradients becomes a bottleneck that significantly drags down the training speed. While for federated learning, the system capacity is usually confined by communication latency between the server and workers. Memory constraints are less studied compared with bandwidth (Sohoni et al., 2019). However, given the trend of an increasing model size in both computer vision and natural language processing, e.g., ViT (0.6B) (Dosovitskiy et al., 2020), Megatron-LM (8.3B) (Shoeybi et al., 2019), T5 (11B) (Raffel et al., 2019), it evidences that memory is becoming an unavoidable constraint for training large models in current GPU generation with 16/32GB memory. For federated learning, it is also necessary to reduce the memory footprint on workers especially for those IoT applications.

In this work, we will focus on the *parsimonious* setting, jointly dealing with limited bandwidth and memory. The priority is still communication efficiency for speeding up training time, otherwise one can simply rely on SGD with almost no additional memory consumption. Many existing works are unable to cope with such a challenging problem because memory is not carefully taken care of.

**Communication efficiency.** A well-documented approach to alleviate communication bottleneck is to compress the (stochastic) gradients such that training can be made faster with reduced overhead (Seide et al., 2014; Alistarh et al., 2017; Karimireddy et al., 2019). Depending on whether the gradient compressor is biased, these schemes are categorized as follows.

*Unbiased compressors* maintain the unbiasedness of stochastic gradients at the price of enlarged variance. For example, uniform quantization of stochastic gradients is studied in QSGD (Alistarh et al., 2017). A concurrent work (Wen et al., 2017) focuses on a special case of QSGD by quantizing each entry of the gradient into $\{\pm 1, 0\}$. Other variants include natural quantization (Horvath et al., 2019), non-uniform quantization (Ramezani-Kebrya et al., 2021) and adaptive quantization (Faghri et al., 2020). Another route to obtain an unbiased compressor is through (scaled) gradient sparsification (Wangni et al., 2018) or its generalized form, atomic decomposition (Wang et al., 2018). However, such methods may be practically slow with performance degradation (Vogels et al., 2019).

*Biased gradient compressors*, on the other hand, have been more successful in practice since not only do they support a more impressive compression ratio, but also a test accuracy comparable with SGD can be obtained in many scenarios. Examples of such compressors contain top-$k$ or (unscaled) random-$k$ sparsification (Lin et al., 2018; Alistarh et al., 2019), signSGD (Bernstein et al., 2018a;b), and PowerSGD (Vogels et al., 2019). Due to the bias, such compressors typically rely on error feedback (EF) schemes to ensure convergence (Stich et al., 2018; Karimireddy et al., 2019).

**Memory concerns in communication efficient methods.** Memory concerns are highly entangled with communication efficiency in distributed training. On the one hand, memory footprint can be alleviated through a smaller batchsize (Krause et al., 2016; Spring et al., 2019), leading to increased iterations as well as communication rounds per epoch. Hence, communication efficient methods are particularly useful for accelerating training time in such cases. On the other hand, existing communication efficient methods are challenged by limited memory due to several reasons. First, a smaller batch size, which enlarges the variance, degrades the applicability of unbiased gradient compressors. Confirmed in our experiments, the exploded variance typically calls for a small step size which decelerates convergence. Unbiased gradient compressors such as quantization, are also hindered by the need of AllGather (Xu et al., 2020), since peak memory proportional to the number of workers is required. This renders them inefficient for the parsimonious setting, especially when the number of workers is large. Memory is also not handled well in methods with biased gradient compressors due to the need of error feedback, where additional memory the same as model size has to be consumed to keep track of accumulated compression error. Note that although no memory footprint is introduced in Local SGD (Stich, 2019), it suffers from variance blown problem as well. More importantly, local SGD cannot be integrated with ZeRO 3 (Rajbhandari et al., 2020), state-of-the-art method for coping with limited memory, due to partitioned optimizer states among workers.

In this work, we will focus on EF-based methods because of the capability of adopting a biased gradient compressor, which is typically contractive thus more robust to the gradient variance in a small batchsize setting. The additional memory consumption of EFSGD over SGD is the local error vector, which is delicate and critical for ensuring convergence. Meanwhile, the designed manner for memory efficiency can only come with lightweight and negligible runtime overhead, otherwise it violates the ultimate goal of communication efficient methods. Contractive error feedback is thus introduced to address these challenges. In a nutshell, our contributions can be summarized as:

- To the best of our knowledge, this is the first work that systematically studies memory concerns in communication efficient methods. Contractive error feedback (ConEF), is introduced to effectively manage memory given the preference of an allreducable and biased gradient compressor.

- Convergence of ConEF is established. Our theoretical results suggest a tradeoff between memory efficiency and a faster convergence, and ConEF finds the sweet spot.

- ConEF is capable of saving $80\% - 90\%$ additional memory of EFSGD on various learning problems such as image classification, language modeling, and machine translation. With almost the same runtime of EFSGD, ConEF achieves 1.3x – 5x speedup over SGD. Though test performance slightly drops on smaller models, the proposed method is more useful and reliable for larger networks where improved test accuracy over EFSGD is observed.

**Notation**. Bold lowercase (uppercase) letters denote vectors (matrices); $\|\mathbf{x}\|$ stands for $\ell_2$ norm of $\mathbf{x}$; and $\langle \mathbf{x}, \mathbf{y} \rangle$ denotes the inner product of $\mathbf{x}$ and $\mathbf{y}$. In addition, we use $[\mathbf{x}]_i$ ($[\mathbf{X}]_{i,j}$) to denote the $i$-th entry of vector $\mathbf{x}$ ($i, j$-th entry of matrix $\mathbf{X}$).

## 2 PRELIMINARIES

This section briefly recaps EFSGD (Stich et al., 2018; Karimireddy et al., 2019) listed under Alg. 1. We use $\mathbf{g}_t^i$ to denote the stochastic gradient at $\mathbf{x}_t$ on worker $i$, and assume that the stochastic gradients are mutually independent among different workers per iteration. In line 5, the scaled

| **Algorithm 1** EFSGD | **Algorithm 2** ConEF |
|---|---|
| 1: **Initialize:** $\mathbf{x}_0 \in \mathbb{R}^d, \mathbf{e}_0^i = \mathbf{0} \in \mathbb{R}^d, \forall i, \eta$ | 1: **Initialize:** $\mathbf{x}_0 \in \mathbb{R}^d, \mathbf{e}_0^i = \mathbf{0} \in \mathbb{R}^d, \forall i, \eta$ |
| 2: **for** $t = 0, 1, \ldots, T - 1$ **do** | 2: **for** $t = 0, 1, \ldots, T - 1$ **do** |
| 3:      assert $\mathbf{x}_t = \mathbf{x}_t^i$ for every worker $i$ | 3:      assert $\mathbf{x}_t = \mathbf{x}_t^i$ for every worker $i$ |
| 4:      **for** worker $i = 1, \ldots, N$ in parallel **do** | 4:      **for** worker $i = 1, \ldots, N$ in parallel **do** |
| 5:          $\mathbf{p}_t^i = \eta \mathbf{g}_t^i + \mathbf{e}_t^i$ | 5:          $\mathbf{p}_t^i = \eta \mathbf{g}_t^i + \mathbf{e}_t^i$ |
| 6:          $\boldsymbol{\Delta}_t^i = \mathcal{Q}(\mathbf{p}_t^i)$ | 6:          $\boldsymbol{\Delta}_t^i = \mathcal{Q}(\mathbf{p}_t^i)$ |
| 7:          $\boldsymbol{\Delta}_t = \mathrm{Aggregate}(\boldsymbol{\Delta}_t^i, \forall i)$ | 7:          $\boldsymbol{\Delta}_t = \mathrm{Aggregate}(\boldsymbol{\Delta}_t^i, \forall i)$ |
| 8:          $\mathbf{x}_{t+1} = \mathbf{x}_t - \boldsymbol{\Delta}_t$ | 8:          $\mathbf{x}_{t+1} = \mathbf{x}_t - \boldsymbol{\Delta}_t$ |
| 9:          $\mathbf{e}_{t+1}^i = \mathbf{p}_t^i - \boldsymbol{\Delta}_t^i$ | 9:          $\mathbf{e}_{t+1}^i = \mathcal{C}(\mathbf{p}_t^i - \boldsymbol{\Delta}_t^i)$ |
| 10:      **end for** | 10:      **end for** |
| 11: **end for** | 11: **end for** |

stochastic gradient is augmented via accumulated compression error $\mathbf{e}_t^i$, then compressed and communicated. There is no restriction to the biasedness of gradient compressor $\mathcal{Q}$. Examples of $\mathcal{Q}$ include (scaled) sign, random/top-$k$, and powerSGD. The "aggregate" in line 7 refers to $\boldsymbol{\Delta}_t = \frac{1}{N} \sum_i \boldsymbol{\Delta}_t^i$, and the compression error $\mathbf{e}_{t+1}^i$ is recomputed in line 9 after updating model $\mathbf{x}_t$. More implementation details on communication of gradients, and reasons for the preference of AllReduce, are deferred to Apdx. A.1 to save space.

Besides the widely appreciated allreducable gradient compressors, those biased ones are more tailored for settings where the inclination is to rely on a smaller batchsize for memory saving. This is because biased gradient compressors are usually contractive; see more details shortly in Assumption 4. However, due to the need of error feedback to ensure convergence, the additional memory consumption of $\mathbf{e}_t^i$ can confine the applicability of biased compressors as well as EFSGD in parsimonious setups, and this issue is somehow omitted by existing works. Other error feedback variants, e.g., (Wu et al., 2018; Basu et al., 2019; Xu et al., 2021; Richtárik et al., 2021), also rely on additional vectors for the compression error, hence may benefit from the proposed technique as well.

## 3 MEMORY SAVING VIA CONTRACTIVE ERROR FEEDBACK

To endow EFSGD with memory efficiency, contractive error feedback is introduced in this section. The key idea is to apply another compressor $\mathcal{C}$ on the error vector $\mathbf{e}_t^i$ such that the memory footprint can be mitigated. The proposed method is summarized in Alg. 2, where the name contractive error feedback (ConEF) originates from the fact that the error vector is represented in its compressed format. Note that the encoding and decoding of compression in Alg. 2 are omitted for notational convenience. $\mathcal{Q}$ and $\mathcal{C}$ are adopted to denote the gradient and error compressors, respectively, to highlight their different roles. The gradient compressor $\mathcal{Q}$ is general and a biased one is more recommended for small batchsizes that can squeeze more memory out. An unbiased error compressor $\mathcal{C}$ is preferable since it is helpful for generalization as we shall discuss later in Section 4. Moreover, the implementation of $\mathcal{C}$ has to be lightweight to avoid slowing down runtime. We will focus on theoretical properties first, and practical guidances for the choice of $\mathcal{C}$ are deferred to Section 4.

### 3.1 CONVERGENCE

Convergence of Alg. 2 is established in this subsection under standard assumptions for communication efficient algorithms in the distributed setting (Alistarh et al., 2017; Karimireddy et al., 2019; Stich et al., 2018; Zheng et al., 2019; Abdi & Fekri, 2020; Basu et al., 2019; Alistarh et al., 2019).

**Assumption 1.** *The objective function* $f : \mathbb{R}^d \to \mathbb{R}$ *has L-Lipchitz continuous gradients; that is,* $\|\nabla f(\mathbf{x}) - \nabla f(\mathbf{y})\| \le L \|\mathbf{x} - \mathbf{y}\|, \forall \mathbf{x}, \mathbf{y} \in \mathbb{R}^d.$

**Assumption 2.** *The stochastic gradient* $\mathbf{g}_t^i$ *is unbiased with bounded variance* $\mathbb{E}[\mathbf{g}_t^i | \mathbf{x}_t] = \nabla f(\mathbf{x}_t)$, *and* $\mathbb{E}[\|\mathbf{g}_t^i - \nabla f(\mathbf{x}_t)\|^2 | \mathbf{x}] \le \sigma^2$. *The gradient is also upper bounded* $\mathbb{E}[\|\nabla f(\mathbf{x}_t)\|^2] \le G^2$.

**Assumption 3.** *(Unbiased error compressor* $\mathcal{C}$*). For any given* $\mathbf{x}$, $\mathbb{E}[\mathcal{C}(\mathbf{x}) | \mathbf{x}] = \mathbf{x}$. *In addition, there exist* $\theta \ge 0$ *such that* $\mathbb{E}[\|\mathcal{C}(\mathbf{x}) - \mathbf{x}\|^2 | \mathbf{x}] \le \theta \|\mathbf{x}\|^2$.

**Assumption 4.** *(Contractive gradient compressor* $\mathcal{Q}$*). For any given* $\mathbf{x}$, *there exists* $\delta \in (0, 1)$ *such that* $\mathbb{E}[\|\mathcal{Q}(\mathbf{x}) - \mathbf{x}\|^2 | \mathbf{x}] \le \delta \|\mathbf{x}\|^2$.

Holding true for many allreducable and biased gradient compressors such as unscaled random-(block)-$k$ and powerSGD, Assumption 4 implies contractivity – the compression error is always smaller than the norm square of the compressed vector. Such compressors are more preferable when working with small batchsizes (for memory saving) since they do not amplify the gradient variance

as explained in Apdx. A.2 and (Stich et al., 2018). Unbiased compressors satisfying Assumption 4 are usually not allreducable; see also Apdx. A.2. The convergence of Alg. 2 is established below.

**Theorem 1.** *Suppose that Assumptions 1 - 4 hold and $T$ is large. Choosing $\eta = \mathcal{O}\big(\frac{1}{L(\sqrt{T/N} + T^{1/3})}\big)$, Alg. 2 guarantees that*

$$\frac{1}{T} \sum_{t=0}^{T-1} \mathbb{E}\big[\|\nabla f(\mathbf{x}_t)\|^2\big] \leq \mathcal{O}\bigg( \frac{L\big(f(\mathbf{x}_0) - f^*\big) + (\sigma^2 + G^2)}{\sqrt{NT}} \bigg) + \mathcal{O}\bigg( \frac{\delta\theta(\sigma^2 + G^2)}{\sqrt{NT}} \bigg). \quad (2)$$

The dependence on $N$ demonstrates that linear speedup is achieved by ConEF. The second term in the right hand side of (2) is the additional error term incurred by compression. Next, we compare it with existing works to gain deeper understanding of ConEF. Upon relating Theorem 1 with EFSGD (Karimireddy et al., 2019), the price paid for memory efficiency can be clearly visualized. Since only the single worker case is considered in the EFSGD paper, we let $N = 1$ in Theorem 1 for a fair comparison. From the additional error term of EFSGD, i.e., $\mathcal{O}(\frac{\delta(\sigma^2 + G^2)}{T})$, it is observed that ConEF converges slower. However, ConEF still benefits from the locally compressed error vector when further compared to QSGD which does not employ error feedback schemes (Alistarh et al., 2017). Suppose that $\mathcal{Q}$ is unbiased with Assumption 3 satisfied, then the error term in QSGD is $\mathcal{O}\big(\frac{(1+\theta)(\sigma^2 + G^2)}{\sqrt{NT}}\big)$. Therefore, ConEF improves the dependence on $\theta$ over QSGD. In addition, Theorem 1 is also tighter than (Horváth & Richtárik, 2020) since their error term is $\mathcal{O}(\frac{(1+\delta\theta)(\sigma^2 + G^2)}{\sqrt{NT}})$.

**ConEF finds the sweet spot.** The comparisons above suggest a tradeoff between memory efficiency and a fast convergence. QSGD is at one extreme where no extra memory is needed at the price of slower convergence due to the blown variance introduced by unbiased gradient compressors. Note also that in general biased gradient compressors cannot be applied in QSGD. EFSGD is at another extreme, where the extra memory consumption is induced by $\mathbf{e}_t^i$ with the benefit of a small error term in the convergence rate. Our proposed Alg. 2 lives in between EFSGD and QSGD, it saves memory compared to EFSGD, but incurs an error term smaller than QSGD yet larger than EFSGD.

### 3.2 IMPROVED CONEF

In this subsection, we show that when $\mathcal{Q}$ (or $\mathcal{C}$) is accurate, the dependence on $\theta$ (resp. $\delta$) in Theorem 1 can be improved. The basic idea is to apply contractive error feedback on top of contractive error feedback such that the compression error can be further reduced. The resultant algorithm, improved ConEF (iConEF), is summarized in Alg. 3. Depending on whether $\mathcal{C}$ or $\mathcal{Q}$ is used twice, there are two versions of iConEF as marked in lines 9 – 13.

Equipping with an accurate $\mathcal{C}$ with $\theta < 1$, the compression error can be reused to retain performance. Mathematically, this error $\mathbf{p}_t^i - \mathbf{\Delta}_t^i - \tilde{\mathbf{e}}_{t+1}^i$ is once again compressed and added back to $\mathbf{p}_{t+1}^i$ in the next iteration to reduce the variance of $(\tilde{\mathbf{e}}_{t+1}^i + \mathbf{q}_{t+1}^i)$; see (20). However, this approach might require

---

**Algorithm 3** iConEF

1: **Initialize:** $\mathbf{x}_0, \tilde{\mathbf{e}}_0^i = \mathbf{0}, \mathbf{q}_0^i = \mathbf{0}, \eta$
2: **for** $t = 0, 1, \ldots, T - 1$ **do**
3:     assert $\mathbf{x}_t = \mathbf{x}_t^i$ for every worker $i$
4:     **for** worker $i = 1, \ldots, N$ in parallel **do**
5:         $\mathbf{p}_t^i = \eta \mathbf{g}_t^i + \tilde{\mathbf{e}}_t^i + \mathbf{q}_t^i$
6:         $\mathbf{\Delta}_t^i = \mathcal{Q}(\mathbf{p}_t^i)$
7:         $\mathbf{\Delta}_t = \text{Aggregate}(\mathbf{\Delta}_t^i, \forall i)$
8:         $\mathbf{x}_{t+1} = \mathbf{x}_t - \mathbf{\Delta}_t$
9:         **if** version 1 **then:**
10:           $\tilde{\mathbf{e}}_{t+1}^i = \mathcal{C}(\mathbf{p}_t^i - \mathbf{\Delta}_t^i)$    ▷ iConEF-v1
11:         **else:**
12:           $\tilde{\mathbf{e}}_{t+1}^i = \mathcal{Q}(\mathbf{p}_t^i - \mathbf{\Delta}_t^i)$    ▷ iConEF-v2
13:         **end if**
14:         $\mathbf{q}_{t+1}^i = \mathcal{C}(\mathbf{p}_t^i - \mathbf{\Delta}_t^i - \tilde{\mathbf{e}}_{t+1}^i)$
15:     **end for**
16: **end for**

---

to store both $\tilde{\mathbf{e}}_{t+1}^i$ and $\mathbf{q}_{t+1}^i$, thus less efficient in terms of memory consumption compared to Alg. 2 with the exception of cases where $\mathcal{C}$ is a linear compressor such as count sketch.

**Theorem 2.** *Suppose that Assumption 3 holds with $0 \leq \theta < 1$. Given Assumptions 1, 2 and 4, and choosing $\eta = \mathcal{O}\big(\frac{1}{L(\sqrt{T/N} + T^{1/3})}\big)$, iConEF-v1 guarantees that*

$$\frac{1}{T} \sum_{t=0}^{T-1} \mathbb{E}\big[\|\nabla f(\mathbf{x}_t)\|^2\big] \leq \mathcal{O}\bigg( \frac{L\big(f(\mathbf{x}_0) - f^*\big) + (\sigma^2 + G^2)}{\sqrt{NT}} \bigg) + \mathcal{O}\bigg( \frac{\delta\theta^2(\sigma^2 + G^2)}{\sqrt{NT}} \bigg).$$

The $\theta^2$ in the last term tightens Theorem 1 showcasing the benefit of the third compressor. Examples of an accurate $\mathcal{C}$ satisfying $0 \leq \theta < 1$ include quantization working in "dense regime" (Alistarh

et al., 2017). In particular, when the quantization level is chosen as $s = 2\sqrt{d}$, we have $\theta = \frac{1}{4}$. Note also that it is possible to encode a compressed vector with less than $3.44d + 32$ bits (Alistarh et al., 2017, Lemma A.6). This means the memory reduction is almost $90\%$. Another example is natural compression that can offer at least 3.5x memory saving with $\theta \le \frac{1}{8}$ (Horvath et al., 2019).

Similarly, when $\mathcal{Q}$ satisfies Assumption 4, which amounts to most of commonly adopted biased gradient compressors with $\delta < 1$, the dependence on $\delta$ can be improved through iConEF-v2.

**Theorem 3.** *Suppose that Assumptions 1 – 4 hold. Choosing $\eta = \mathcal{O}\big(\frac{1}{L(\sqrt{T/N}+T^{1/3})}\big)$, iConEF-v2 guarantees that*

$$\frac{1}{T}\sum_{t=0}^{T-1}\mathbb{E}\big[\|\nabla f(\mathbf{x}_t)\|^2\big] \le \mathcal{O}\left(\frac{L\big(f(\mathbf{x}_0)-f^*\big)+(\sigma^2+G^2)}{\sqrt{NT}}\right) + \mathcal{O}\left(\frac{\delta^2\theta(\sigma^2+G^2)}{\sqrt{NT}}\right).$$

## 4 A CLOSER LOOK AT ERROR COMPRESSORS

Previous sections have coped with contractive error feedback in its general form. In what follows, we will discuss the error compressor $\mathcal{C}$ in detail to gain more insights and practical guidances.

### 4.1 GENERALIZATION PROPERTIES

The merits of an unbiased $\mathcal{C}$ over a biased one come from generalization properties. To see this point, consider overparameterized least square problems (Wilson et al., 2017) as an example

$$\min_{\mathbf{x}} f(\mathbf{x}) := \frac{1}{2}\|\mathbf{A}\mathbf{x} - \mathbf{y}\|^2 \tag{3}$$

where $\mathbf{A} \in \mathbb{R}^{n \times d}$ with $d \gg n$ is the data matrix and $\mathbf{y} \in \mathbb{R}^d$ denotes the labels. Since $d$ is larger than the number of data samples $n$, this overparameterized problem has multiple global minima with zero loss collected in the set $\mathcal{X}^* := \{\mathbf{x}|f(\mathbf{x}) = 0\}$. Define $\mathcal{R}(\mathbf{X})$ as the range of $\mathbf{X}$. Given the fact that any stochastic gradient of (3) lies in $\mathcal{R}(\mathbf{A}^\top)$, it is straightforward that SGD converges to a point in $\mathcal{R}(\mathbf{A}^\top)$ when initialized well. It is further shown that the solution SGD converges to has the smallest norm among $\mathcal{X}^*$, i.e., $\mathbf{x}^* = \arg\min_{\mathbf{x} \in \mathcal{X}^*} = \mathbf{A}^\top(\mathbf{A}\mathbf{A}^\top)^{-1}\mathbf{y}$. This $\mathbf{x}^*$ is also known as the maximum margin solution, which leads to potential generalization benefits (Valiant, 1984; Cortes & Vapnik, 1995). In fact, for problem (3), if the iterates of a converging algorithm lie in $\mathcal{R}(\mathbf{A}^\top)$, finding the maximum margin solution is ensured (Karimireddy et al., 2019, Lemma 9).

EFSGD converges to a point that is sufficiently close to $\mathcal{R}(\mathbf{A}^\top)$ (Karimireddy et al., 2019). This gives the potential generalization benefit when adopting biased gradient compressors $\mathcal{Q}$ compared with e.g., sign-SGD (Bernstein et al., 2018a;b). Next, we show that such a generalization merit is maintained in Alg. 2 and its variants when $\mathcal{C}$ is unbiased.

**Theorem 4.** *If Assumptions 1 - 4 hold, initialized at $\mathbf{x}_0 \in \mathcal{R}(\mathbf{A}^\top)$, ConEF ensures $\mathbb{E}\big[\mathbf{x}_t - \frac{1}{N}\sum_i \mathbf{e}_t^i\big] \in \mathcal{R}(\mathbf{A}^\top)$; moreover, iConEF-v1 and v2 guarantee $\mathbb{E}\big[\mathbf{x}_t - \frac{1}{N}\sum_i \mathbf{q}_t^i - \frac{1}{N}\sum_i \tilde{\mathbf{e}}_t^i\big] \in \mathcal{R}(\mathbf{A}^\top)$.*

Note that for ConEF, we have $\|\mathbb{E}[\frac{1}{N}\sum_{i=1}^{N}\mathbf{e}_t^i]\|^2 \le \mathbb{E}[\|\frac{1}{N}\sum_{i=1}^{N}\mathbf{e}_t^i\|^2] = \mathcal{O}(\eta^2)$. This implies that $\mathbf{x}_t$ is close to the maximum margin solution at convergence. We are unable to obtain such a property for a biased $\mathcal{C}$. The potential generalization merits justify the unbiased choice for $\mathcal{C}$.

### 4.2 CHOICES OF ERROR COMPRESSOR

Recall that a prudent error compressor $\mathcal{C}$ for ConEF should: i) satisfy Assumption 3; and, ii) have a lightweight implementation to avoid runtime overhead. Two representative examples, quantization and count sketch, will be discussed in detail in this subsection.

#### 4.2.1 QUANTIZATION

Quantization is a large family of compression schemes. For the ease of presentation, here we only describe the basic version (Alistarh et al., 2017). Given $s$ uniformly distributed quantization levels in $[0, 1]$, this approach quantizes the $i$-the entry of $\mathbf{e}$ as

$$\mathcal{C}([\mathbf{e}]_i) = \|\mathbf{e}\| \cdot \text{sign}([\mathbf{e}]_i) \cdot \xi_i^s([\mathbf{e}]_i) \tag{4}$$

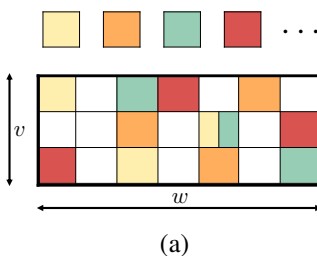
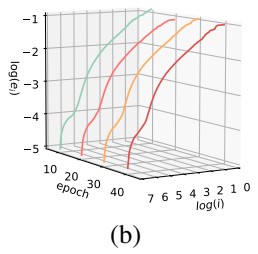
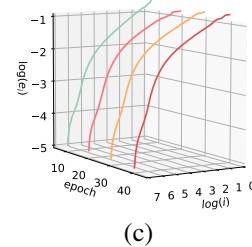

|     |     |     |
|:---:|:---:|:---:|
| (a) | (b) | (c) |

Figure 1: (a) An illustration of count sketch compressing the first 4 entries of $\mathbf{e}_t^i$. A collision is at $[\mathbf{S}]_{2,5}$ (index starts with 1); (b) magnitude of $\mathbf{e}_t^i$ in EFSGD; (c) magnitude of $\mathbf{e}_t^i$ in partial EFSGD.

where $\xi_i^s(\cdot)$ is a random variable designed to ensure the unbiasedness of $\mathcal{C}([\mathbf{e}]_i)$. In particular, let integer $l \in [0, s]$ such that $\frac{|[\mathbf{e}]_i|}{\|\mathbf{e}\|} \in [\frac{l}{s}, \frac{l+1}{s}]$, and then $\xi_i^s([\mathbf{e}]_i)$ is set to $\frac{l+1}{s}$ with probability $\frac{|[\mathbf{e}]_i|}{\|\mathbf{e}\|}s - l$ and $\frac{l}{s}$ otherwise. Assumption 3 is naturally satisfied according to the proposition below.

**Proposition 1.** *(Alistarh et al., 2017) For any* $\mathbf{e}$*, the quantization in* (4) *satisfies: i)* $\mathbb{E}[\mathcal{C}(\mathbf{e})] = \mathbf{e}$*; ii)* $\mathbb{E}[\|\mathcal{C}(\mathbf{e}) - \mathbf{e}\|^2] \leq \min\{\frac{d}{s^2}, \frac{\sqrt{d}}{s}\}\|\mathbf{e}\|^2$*; and, iii) there exists an encoding scheme that requires* $\tilde{\mathcal{O}}(s^2 + s\sqrt{d})$ *bits in expectation to represent* $\mathcal{C}(\mathbf{e})$*.*

When setting quantization level $s = \mathcal{O}(1)$, the memory saving is impressive since it only requires $\mathcal{O}(\sqrt{d})$ bits for a $d$-dimensional vector at the cost of additional implementation burden of e.g., Elias encoding. Stochastic quantization is also suitable for the error compressor $\mathcal{C}$ in iConEF-v1 since it can ensure $\theta < 1$ when $s = \mathcal{O}(\sqrt{d})$.

Other quantization approaches can also be applied as the error compressor. For instance, one can use int-quantization (Mishchenko et al., 2021), where a fp32 number is compressed into int8 or int16; and non-uniform quantization (Ramezani-Kebrya et al., 2021) is also a valid choice.

### 4.2.2 COUNT SKETCH

While quantization is straightforward to come up with, it relies on bit operations for compression rather than working on float numbers directly. Therefore, we introduce count sketch (Charikar et al., 2002) in this subsection, whose implementation is simply incrementing entries in a matrix.

Count sketch compresses a vector $\mathbf{e}$ by maintaining a matrix $\mathbf{S}$ of size $v \times w < d$, i.e., $\mathcal{C}(\mathbf{e}) := \mathbf{S}$. Both $v$ and $w$ are chosen according to some desirable accuracy guarantees discussed shortly. Count sketch relies on hash functions $h_i : \{1, 2, \ldots, d\} \mapsto \{1, 2, \ldots, w\}$ and random sign functions $s_i : \{1, 2, \ldots, d\} \mapsto \{+1, -1\}$ in $i$-th row of $\mathbf{S}$. Count sketches leverage $[\mathbf{e}]_p$ to update each row $i$ of $\mathbf{S}$ by incrementing $h_i(p)$-th column by $s_i(p)[\mathbf{e}]_p$. Compression of $\mathbf{e}$ can be obtained by updating $\mathbf{S}$ with $\{(p, [\mathbf{e}]_p)\}_{p=1}^d$; see Fig. 1 (a). To decompress and estimate $[\mathbf{e}]_p$ from $\mathbf{S}$, one needs to calculate $\text{Median}(s_i(p) \cdot [\mathbf{S}]_{i,h_i(p)}, \forall i)$. Replacing median with mean also works, but typically median performs better. Count sketch satisfies Assumption 3 as shown in the following proposition.

**Proposition 2.** *(Charikar et al., 2002) Count sketch is unbiased, i.e.,* $\mathbb{E}[\mathcal{C}(\mathbf{e})] = \mathbf{e}$*. With* $w = \Theta(\frac{1}{\epsilon^2})$ *and* $v = \Theta(\log \frac{d}{p})$*, count sketch ensures* $\|\mathbf{e} - \mathcal{C}(\mathbf{e})\|_\infty \leq \epsilon\|\mathbf{e}\|$ *with probability at least* $1 - p$*.*

Proposition 2 (Spring et al., 2019) suggests that count sketch is unbiased. Using inequality $\|\mathbf{a}\|_\infty \leq \|\mathbf{a}\| \leq \sqrt{d}\|\mathbf{a}\|_\infty$, it is not difficult to verify $\|\mathbf{e} - \mathcal{C}(\mathbf{e})\| \leq \epsilon\sqrt{d}\|\mathbf{e}\|$. Although count sketch satisfies a high probability variant of Assumption 3 with $\theta = \epsilon^2 d$, it works well in our experimental studies with the recommendation of $v = 1$ or $v = 3$. Hence, this error compressor is more practical relative to the theoretical bounds. Another useful property of count sketch is the linearity, i.e., $\mathcal{C}(\mathbf{e}_1 + \mathbf{e}_2) = \mathcal{C}(\mathbf{e}_1) + \mathcal{C}(\mathbf{e}_2)$. This is helpful since i) it opens the possibility to squeeze $\tilde{\mathbf{e}}_{t+1}$ and $\mathbf{q}_{t+1}$ in iConEF-v1 to further reduce the memory consumption by adding these two count sketches; and ii) it eases error reset (Basu et al., 2019; Xie et al., 2020), which synchronizes the local errors every a few iterations among all workers. The second merit is because count sketch is small in size and allreducable, and more on this can be found in Apdx. G.4.

Empirically, count sketch performs best when the compressed vector follows power law[1]. We train ResNet18 on CIFAR10, and plot the magnitude of $\mathbf{e}_t^i$ on the first worker at the end of epochs

---

[1]The magnitude-index plot is roughly a line in log-log scale.

Table 1: $\beta$ vs. sketch size for convergence

| $\beta$ | 0.1 | 0.2 | 0.3 | 0.4 | 0.5 | 0.6 | 0.7 | 0.8 | 0.9 |
|---|---|---|---|---|---|---|---|---|---|
| smallest sketch size | 0.9 | 0.8 | 0.7 | 0.6 | 0.4 | 0.4 | 0.3 | 0.2 | 0.1 |

$\{10, 20, 30, 40\}$ in Fig. 1(b). It is observed that power law is not satisfied. However, in Fig. 1(c) we find that the error vectors in a variant of EFSGD, termed partial EFSGD (Abdi & Fekri, 2020) (see Alg. 4 in Apdx. F due to space limitation), follow broken power law, that is, a piece-wise linear function in the log-log plot. Partial EFSGD adds part of the compression error, i.e., $(1 - \beta)\mathbf{e}_t^i$ for $\beta \in [0, 1)$, to stochastic gradients before compressing. As we shall see in numerical experiments, the update in partial EFSGD is helpful to save more memory when working with count sketch.

## 5 NUMERICAL RESULTS

In this section, experimental studies are carried out to validate the proposed algorithms. Since our major goal is to push the limit of memory saving, we will mostly focus on ConEF, and experiments of iConEF can be found in Apdx. G.4. Overall, the experiments are designed to answer three research questions (RQs): 1) how should we choose hyperparameters in ConEF. Since quantization is well-known in community, we put more efforts on the less understood count sketch; 2) when the gradient is aggressively compressed, how much memory can be saved; and, 3) how will the proposed method perform on different learning tasks. Although allreducable gradient compressors are advocated, we also include numerical results for scaled-sign, which requires AllGather for communication, to mimic parameter-server scenarios in RQ1. Testing various gradient compressors is also helpful to demonstrate the generality of ConEF. We follow common practice where both gradient and error compressors are applied in their layerwise form. Our implementation utilizes `torch.DistributedDataParallel` so that the bucket trick can be leveraged to overlap gradient computation and communication, and NCCL is chosen as communication backend.

### 5.1 RQ1: GUIDANCE ON HYPERPARAMETER CHOICE

In this subsection, we focus on the parameter choices of count sketch and quantization in ConEF using scaled-sign gradient compressor $\mathcal{Q}(\mathbf{g}) = \frac{\|\mathbf{g}\|_1}{d}\text{sign}(\mathbf{g})$ as an example. About 97% communication overhead can be reduced. As AllGather is required for communication, the peak memory consumption is increased. Such a gradient compressor is adopted to simulate the parameter-server setting, where there is enough memory capacity on the server side. We test this setting with 4 GPUs on an Amazon EC2 p3.8xlarge instance, where more implementation details are put into Apdx. G.1 given the page limitation. When using allreducable gradient compressors, the general guidance for parameter choices is similar, although the numbers might change; see Apdx. G.1 for additional illustration of this point. Most of experiments are tested on CIFAR10,[2] which consists of 60000 32x32 images in 10 classes with 50000 training samples and 10000 test ones. Standard data augmentation and preprocessing techniques are employed (He et al., 2016).

We start with comparing contractive error feedback for vanilla and partial EFSGD using count sketches. With $v = 1$, the sketch size has to be 0.95x model size to ensure a good convergence for compressing the error vector in vanilla EFSGD. On the other hand, the error vector of partial EFSGD follows the broken power law, and hence is more suitable for count sketches. We list the choice of $\beta$ and the smallest sketch size for convergence in Table. 1. Note that only $\{0.9, 0.8, 0.7, 0.6, 0.5, 0.4, 0.3, 0.2, 0.1\}$ are tested, though there is no barrier to choose other values. It is observed that a larger $\beta$ leads to more saved memory. This is possibly because that a larger $\beta$ helps to suppress the error accumulation in count sketches; see Apdx. F.

Next we focus on choices for $v$ and $w$ in count sketches. We implement the partial EFSGD variant of ConEF (see Alg. 4 in Apdx. F) with memory budget as 0.6x model size. Two specific sketches, $v = 1, w = 0.6$x model size and $v = 3, w = 0.2$x model size, are tested. It can be seen in Fig. 2 that the first sketch with $v = 1$ performs better. This suggests that a larger $w$, leading to less collision of hash functions, is more numerically beneficial compared with an increased $v$. Similar observations also appear on allreducable compressors. Hnece, $v = 1$ is adopted in most of our experiments.

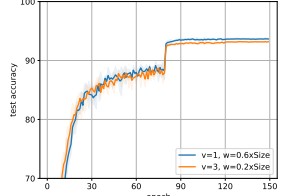

Figure 2: ConEF with different count sketches.

---

[2]https://www.cs.toronto.edu/ kriz/cifar.html

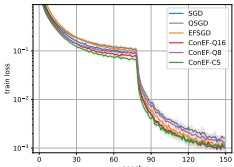 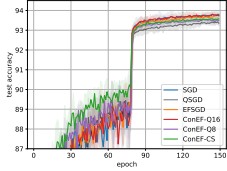 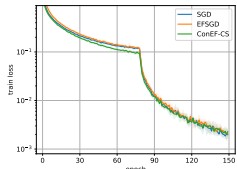 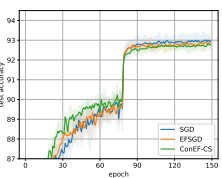

Figure 3: Performance of ConEF on ResNet18 and MobileNetv2. From left to right: (a) train loss and (b) test accuracy on ResNet18; and (c) train loss and (d) test accuracy on MobileNetv2.

After clarifying the parameter choices, the effectiveness of ConEF in parameter-server settings is tested with ResNet18 on CIFAR10 for 150 epochs as shown in Fig. 3. The detailed parameter choices are deferred to Apdx. G.1. We use ConEF-CS, ConEF-Q16 and ConEF-Q8 to denote Alg. 2 with error compressors $\mathcal{C}$ being count sketch, 16-level quantization, and 8-level quantization, respectively. Note that when $\mathcal{C}$ is a stochastic quantizer, we follow common practice to replace the $\ell_2$ norm in (4) with max-norm for superior performance (Alistarh et al., 2017). It is observed that on ResNet18, ConEF-Q16 achieves a test accuracy that is almost the same as SGD and EFSGD, demonstrating EFSGD uses unnecessary memory. On the other hand, 44Mb is saved on a 50Mb model with count sketch, while only incurring a slight drop of 0.24 on test accuracy compared with SGD. To further validate the potential of ConEF in smart phone based applications, we test MobileNetv2 (Sandler et al., 2018), a more tailored model for the targeted setting. We set $w = 0.1\text{x}$ model size to save 8Mb on a 14Mb model compared with EFSGD, while almost losing no test accuracy. Another interesting observation is that the convergence of ConEF-CS tends to be faster in the first 80 epochs before decreasing the learning rate. This might be useful in federated learning with IoT devices where each worker may not participate in training for too many iterations.

## 5.2 RQ2: MEMORY SAVING WITH AGGRESSIVELY COMPRESSED GRADIENTS

Next we focus on allreducable gradient compressors. We use 16 GPUs on 4 Amazon EC2 p3.8xlarge instances for experiments.

To investigate the performance of ConEF under aggressively compressed gradients, powerSGD is selected (Vogels et al., 2019). PowerSGD finds a rank-$r$ approximation to the gradient matrix $\mathbf{G}$. It is allreducable and different levels of communication efficiency can be achieved by tuning $r$. This is a challenging setting for ConEF given that i) the performance of the heuristic powerSGD highly depends on $\mathbf{e}_t^i$, and ii) $\mathbf{e}_t^i$ plays a more important role when the gradient is aggressively compressed.

ResNet18 is trained on CIFAR10 with $r = 4$ in powerSGD, which reduces 98.5% communication overhead. Unlike (Vogels et al., 2019), a small batchsize, e.g., 16 per worker, is considered here for reducing memory requirement (Krause et al., 2016). Mixed precision trick is also adopted to facilitate memory saving, where we use fp16 numbers on count sketches instead of fp32 (Micikevicius et al., 2018). Two more benchmarks are compared with. *qSGD* (Vogels et al., 2019): SGD with an unbiased and allreducable gradient compressor that has the same communication overhead as powerSGD. In particular, the gradient matrix $\mathbf{G} \in \mathbb{R}^{n \times m}$ is approximated by $(\mathbf{GU})\mathbf{U}^\top$ for a random matrix $\mathbf{U} \in \mathbb{R}^{m \times r}$. Small-sized matrices $\mathbf{GU}$ and $\mathbf{U}^\top$ are communicated. We term such a scheme as qSGD to distinguish it with QSGD (Alistarh et al., 2017). Another benchmark is *hEFSGD*, which is a heuristic method by keeping the top-$k$ elements of $\mathbf{e}_t^i$ in EFSGD for memory saving.

The results can be found in Fig. 4, where we mark the percentage of saved memory with ConEF in the figure legend. There are several observations. ConEF-CS has a similar runtime as EFSGD, and both are 10% faster than SGD. This demonstrates the benefit of reduced communication overhead in distributed training. ConEF-CS significantly outperforms qSGD, which confirms the necessity of additional memory to enable a good convergence in memory constraint setting with small batchsizes. When 60% memory is saved, ConEF-CS also outperforms hEFSGD by a large margin in terms of both runtime and test accuracy, validating the generalization merits of an unbiased error compressor (cf. Theorem 4). ConEF-CS has a comparable performance with EFSGD, and more saved memory leads to larger accuracy loss.

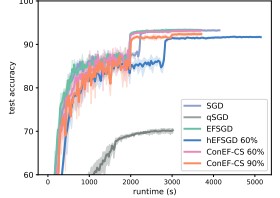

Figure 4: ConEF with a powerSGD gradient compressor.

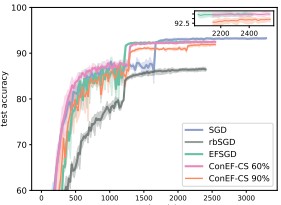 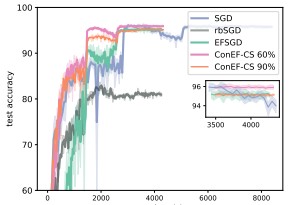 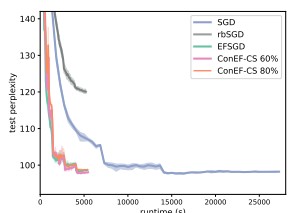

Figure 5: Performance of ConEF with an unscaled random block gradient compressor on: ResNet-18, WideResNet-28-10, and LSTM (left to right).

## 5.3  RQ3: MEMORY SAVING ON VARIOUS LEARNING TASKS

In this subsection, ConEF is tested on different learning tasks. The unscaled random block gradient compressor is adopted, where a continuous block of $k$ coordinates are chosen in a uniformly random fashion to improve $90\%$ communication efficiency. When using the same random seed among GPUs, AllReduce is readily applied. Such a gradient compressor is more suitable for ConEF, since the error vector $\mathbf{e}_t^i$ has $d - k$ zeros. Three different training tasks are considered: i) ResNet-18 on CIFAR10; ii) WideResNet-28-10 on CIFAR10; and iii) LSTM for language modeling on Wikitest-2 (Merity et al., 2016). To demonstrate the necessity of additional memory, an unbiased variant of random block gradient compressor enabled by scaling with importance sampling is also implemented. We term the corresponding method as *rbSGD*. A setting with small batchsizes is still the main focus here, and the results can be found in Fig. 5.

On ResNet-18, it is observed that EFGSD and ConEF save $22\%$ and $21\%$ training time compared with SGD. The percentage of time saving improves over powerSGD even with a smaller compression ratio here. This is because more time is spent on the complicated encoding in powerSGD. It is observed that ConEF significantly outperforms rbSGD, once again demonstrating the need of additional memory when training with a small batchsize of $16$. Compared with EFSGD, ConEF-CS drops $0.04$ and $0.5$ on test accuracy to save $60\%$ and $90\%$ memory, respectively.

For WideResNet-28-10, EFSGD and ConEF are $49.0\%$ and $48.6\%$ faster than SGD. Algorithms with (contractive) error feedback again considerably outperform rbSGD. EFSGD does not catch up with the test accuracy of ConEF-CS with $60\%$ memory saving at a difference of $0.6$, suggesting that EFSGD consumes memory inefficiently. ConEF-CS with $90\%$ memory saving only drops $0.1$ test accuracy compared with EFSGD. In light of the results on ResNet-18, a larger model (WideResNet) appears to be more robust when used jointly with contractive error feedback.

Next we focus on language modeling with LSTM on WikiText-2. All tested algorithms reduce about $78\%$ training time of SGD. rbSGD fails to match the performance of both ConEF-CS and EFSGD. When $60\%$ memory is saved, ConEF-CS has a $0.3$ lower perplexity compared to EFSGD. ConEF-CS with $80\%$ memory saved performs slightly worse than EFSGD with $0.2$ higher test perplexity.

We also train a transformer for machine translation on Multi30K (Elliott et al., 2016) in Apdx. G. Adam based heuristic methods with (contractive) error feedback are adopted. In such a case, even $90\%$ memory saved ConEF-CS has a slightly better performance than EF-Adam.

**Additional experiments.** Performance of iConEF-v1 and v2 can be found in Fig. 6. In this experiment, another allreducable gradient compressor, unscaled random-$k$, is adopted to reduce $90\%$ communication. Both versions of iConEF outperform ConEF, validating our findings in Theorems 2 and 3. Due to space limitation, more implementation details and numerical results regarding random-$k$ gradient compressor are deferred to Apdx. G.4, where we observe that ConEF can reduce $95\%$ additional memory of EFSGD on ResNet18.

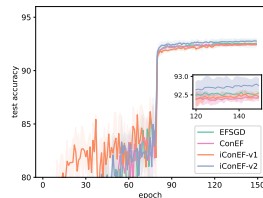

Figure 6: iConEF with a random $k$ gradient compressor.

## 6  CONCLUSION AND FUTURE WORK

Contractive error feedback (ConEF) is introduced to alleviate memory concerns in communication efficient methods. The key insight is that the error vector can be compressed and an effective error compressor, such as count sketch, saves memory with almost no accuracy drop. ConEF is tested on various image classification and language processing tasks, reducing $80\% - 90\%$ of additional memory footprint in EFSGD. Our next step will focus on saving memory more aggressively.

ETHICS STATEMENT

The main topic of this work is speeding up distributed training on devices with limited memory. Since this work focuses on an algorithmic perspective, the impact would be mainly scientific rather than ethical.

REPRODUCIBILITY STATEMENT

All missing proofs can be found in Appendix. Detailed choices for hyperparameters and additional experiments are deferred to Appendix G due to space limitation, and publicly available code will be provided after review process to ensure anonymity.

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

# Supplementary Document

## A  MISSING DETAILS

### A.1  HOW GRADIENTS ARE COMMUNICATED

While the "aggregation" in line 7 of Alg. 1 is a highly abbreviated term, in practice (compressed) gradients are communicated through AllReduce, AllGather, or parameter-server, among which allreducable compressors are widely appreciated. In contrast to AllGather, AllReduce enjoys low communication cost that is nearly independent of the number of workers (Vogels et al., 2019; Peng et al., 2019). In addition, compressed gradients that are allreducable relieve runtime since it only needs to be decoded once. In AllGather, a worker receives $N$ compressed gradients that not only take additional space but also need to be decompressed individually. This has a negative impact on memory and runtime scalability given the number of workers can be large Vogels et al. (2019). When compared to parameter-server, AllReduce bypasses the need of double compression, where compression is applied to the bidirectional communication of clients and server. Parameter-server structure can also introduce communication congestion due to the centralized aggregation at the server side (Yu et al., 2018).

### A.2  BIASED GRADIENT COMPRESSORS ARE MORE SUITABLE FOR SMALL BATCHSIZES

Suppose that $\mathbf{g}$ is a stochastic approximation of $\nabla f(\mathbf{x})$ satisfying Assumption 2. We show that compressors satisfying Assumption 4, typically biased, are more robust to the large variance ($\sigma^2$) in the small batchsize settings. Consider gradient compressor $\mathcal{Q}_1$ satisfying Assumption 4 with $\delta < 1$. The compression error of $\mathcal{Q}_1$ can be calculated as

$$\mathbb{E}[\|\mathcal{Q}_1(\mathbf{g}) - \mathbf{g}\|^2] \leq \delta\mathbb{E}[\|\mathbf{g}\|^2] \leq \delta\mathbb{E}[\|\nabla f(\mathbf{x})\|^2] + \delta\sigma^2. \tag{5}$$

Let $\mathcal{Q}_2$ denote an unbiased compressor satisfying Assumption 3. Typically to get a good compression ratio, we have $\theta > 1$. The compression error is bounded by

$$\mathbb{E}[\|\mathcal{Q}_2(\mathbf{g}) - \mathbf{g}\|^2] \leq \theta\mathbb{E}[\|\mathbf{g}\|^2] \leq \theta\mathbb{E}[\|\nabla f(\mathbf{x})\|^2] + \theta\sigma^2. \tag{6}$$

Given $\theta > 1$, the bound in (6) amplifies the variance; while the bound in (5) shrinks the variance. Therefore, a contractive compressor satisfying Assumption 4, which is typically biased, is more suitable for a small batchsize setting due to the large $\sigma^2$. Although some of unbiased gradient compressors, such as QSGD working in "dense regime" (Alistarh et al., 2017) and natural dithering (Horvath et al., 2019), ensure $\theta < 1$, unfortunately they are not allreducable, and hence are not suitable for our setting.

Other works such as (Stich et al., 2018) also observe the variance blow-up in unbiased gradient compressors.

### A.3  APPLYING CONEF TO ZERO

ZeRO (Rajbhandari et al., 2020) can be used to address memory issues for training large models in a distributed fashion. Mathematically, ZeRO is equivalent to Adam or other chosen optimizers, while the memory issues are addressed by partitioning optimizer states, gradients, and even model parameters among different workers. Here we consider the case of ZeRO3, where all aforementioned parts are partitioned among different workers.

Being most memory efficient, ZeRO3 increases the communication overhead. Hence, it is helpful to use gradient compression to speed up training. However, EF based approach cannot be applied here due to the requirement of a model-sized local error vector that is burdensome to GPU memory. In addition, because of the partitioned gradients, it is impossible to employ the memory trick in (Ramesh et al., 2021) where one puts local error vector to gradient buffer to save space.

We believe ConEF is helpful to reduce the runtime of ZeRO3 while respect constraints on GPUs memory. In this paper, we first demonstrate the feasibility and convergence of ConEF to clear up the theoretical barrier of integrating ConEF to ZeRO3. The detailed implementation is left for future work.

## B    PROOF OF THEOREM 1

*Proof.* For convenience, define the compression error of $\mathcal{C}$ on worker $i$ as $\boldsymbol{\zeta}_t^i = \mathbf{e}_{t+1}^i - \mathbf{p}_t^i + \boldsymbol{\Delta}_t^i$. Following Assumption 3, we have

$$\mathbb{E}\big[\boldsymbol{\zeta}_t^i|\mathbf{p}_t^i, \boldsymbol{\Delta}_t^i\big] = \mathbf{0}, \tag{7}$$

$$\mathbb{E}\big[\|\boldsymbol{\zeta}_t^i\|^2|\mathbf{p}_t^i, \boldsymbol{\Delta}_t^i\big] = \mathbb{E}\big[\|\mathcal{C}(\mathbf{p}_t^i - \boldsymbol{\Delta}_t^i) - \mathbf{p}_t^i + \boldsymbol{\Delta}_t^i\|^2|\mathbf{p}_t^i, \boldsymbol{\Delta}_t^i\big] \leq \theta\|\mathbf{p}_t^i - \boldsymbol{\Delta}_t^i\|^2. \tag{8}$$

Next, we rewrite local error $\mathbf{e}_{t+1}^i$ using $\boldsymbol{\zeta}_t^i$ as

$$\mathbf{e}_{t+1}^i = \mathbf{p}_t^i - \boldsymbol{\Delta}_t^i + \boldsymbol{\zeta}_t^i \overset{(a)}{=} \eta\mathbf{g}_t^i + \mathbf{e}_t^i - \boldsymbol{\Delta}_t^i + \boldsymbol{\zeta}_t^i$$

where (a) follows from $\mathbf{p}_t^i = \eta\mathbf{g}_t^i + \mathbf{e}_t^i$. Then summing over $N$ workers, we have

$$\frac{1}{N}\sum_{i=1}^N \mathbf{e}_{t+1}^i$$

$$= \frac{\eta}{N}\sum_{i=1}^N \mathbf{g}_t^i + \frac{1}{N}\sum_{i=1}^N \mathbf{e}_t^i - \frac{1}{N}\sum_{i=1}^N \boldsymbol{\Delta}_t^i + \frac{1}{N}\sum_{i=1}^N \boldsymbol{\zeta}_t^i$$

$$\overset{(b)}{=} \frac{\eta}{N}\sum_{i=1}^N \mathbf{g}_t^i + \frac{1}{N}\sum_{i=1}^N \mathbf{e}_t^i + \mathbf{x}_{t+1} - \mathbf{x}_t + \frac{1}{N}\sum_{i=1}^N \boldsymbol{\zeta}_t^i$$

where (b) is because $\frac{1}{N}\sum_{i=1}^N \boldsymbol{\Delta}_t^i = \boldsymbol{\Delta}_t = \mathbf{x}_t - \mathbf{x}_{t+1}$. Rearranging the terms and define $\mathbf{z}_t := \mathbf{x}_t - \frac{1}{N}\sum_{i=1}^N \mathbf{e}_t^i$, we arrive at

$$\mathbf{z}_{t+1} = \mathbf{z}_t - \frac{\eta}{N}\sum_{i=1}^N \mathbf{g}_t^i - \frac{1}{N}\sum_{i=1}^N \boldsymbol{\zeta}_t^i. \tag{9}$$

The analysis will rely on (9). By Assumption 1, we have

$$f(\mathbf{z}_{t+1}) - f(\mathbf{z}_t) \tag{10}$$

$$\leq \langle \nabla f(\mathbf{z}_t), \mathbf{z}_{t+1} - \mathbf{z}_t\rangle + \frac{L}{2}\|\mathbf{z}_{t+1} - \mathbf{z}_t\|^2$$

$$= -\frac{1}{N}\sum_{i=1}^N \langle \nabla f(\mathbf{z}_t), \boldsymbol{\zeta}_t^i\rangle - \frac{\eta}{N}\sum_{i=1}^N \langle \nabla f(\mathbf{z}_t), \mathbf{g}_t^i\rangle + \frac{L}{2}\Big\|\frac{\eta}{N}\sum_{i=1}^N \mathbf{g}_t^i + \frac{1}{N}\sum_{i=1}^N \boldsymbol{\zeta}_t^i\Big\|^2$$

$$= -\frac{1}{N}\sum_{i=1}^N \langle \nabla f(\mathbf{z}_t), \boldsymbol{\zeta}_t^i\rangle - \frac{\eta}{N}\sum_{i=1}^N \langle \nabla f(\mathbf{z}_t) - \nabla f(\mathbf{x}_t), \mathbf{g}_t^i\rangle$$

$$- \frac{\eta}{N}\sum_{i=1}^N \langle \nabla f(\mathbf{x}_t), \mathbf{g}_t^i\rangle + \frac{L}{2}\Big\|\frac{\eta}{N}\sum_{i=1}^N \mathbf{g}_t^i + \frac{1}{N}\sum_{i=1}^N \boldsymbol{\zeta}_t^i\Big\|^2.$$

Given (7) and taking expectation w.r.t. the randomness of $\mathcal{Q}$, we arrive at $\mathbb{E}[\boldsymbol{\zeta}_t^i|\mathbf{p}_t^i] = \mathbb{E}[\boldsymbol{\zeta}_t^i|\mathbf{g}_t^i, \mathbf{e}_t^i] = \mathbf{0}$. Then conditioning on $\mathbf{x}_t$ and taking expectation w.r.t. the randomness of $\mathbf{g}_t^i$, we have $\mathbb{E}[\boldsymbol{\zeta}_t^i|\mathbf{x}_t, \mathbf{e}_t^i] = \mathbf{0}$. Hence, we have

$$\mathbb{E}\big[\langle \nabla f(\mathbf{z}_t), \boldsymbol{\zeta}_t^i\rangle|\mathbf{x}_t, \mathbf{e}_t^i\big] = \langle \nabla f(\mathbf{z}_t), \mathbb{E}[\boldsymbol{\zeta}_t^i|\mathbf{x}_t, \mathbf{e}_t^i]\rangle = 0$$

$$\Rightarrow \mathbb{E}\big[\langle \nabla f(\mathbf{z}_t), \boldsymbol{\zeta}_t^i\rangle\big] = 0. \tag{11}$$

We also have

$$\mathbb{E}\big[\langle \nabla f(\mathbf{z}_t) - \nabla f(\mathbf{x}_t), \mathbf{g}_t^i\rangle|\mathbf{x}_t, \mathbf{e}_t^i\big] = \mathbb{E}\big[\langle \nabla f(\mathbf{z}_t) - \nabla f(\mathbf{x}_t), \nabla f(\mathbf{x}_t)\rangle|\mathbf{x}_t, \mathbf{e}_t^i\big].$$

Then taking expectation w.r.t. $\mathbf{x}_t$ and $\mathbf{e}_t^i$, we have

$$-\eta\mathbb{E}\big[\langle\nabla f(\mathbf{z}_t)-\nabla f(\mathbf{x}_t),\mathbf{g}_t^i\rangle\big] = -\eta\mathbb{E}\big[\langle\nabla f(\mathbf{z}_t)-\nabla f(\mathbf{x}_t),\nabla f(\mathbf{x}_t)\rangle\big] \qquad (12)$$

$$\overset{(c)}{\leq} \frac{\eta}{2}\mathbb{E}\big[\|\nabla f(\mathbf{x}_t)\|^2\big]+\frac{\eta}{2}\mathbb{E}\big[\|\nabla f(\mathbf{z}_t)-\nabla f(\mathbf{x}_t)\|^2\big]$$

$$\overset{(d)}{\leq} \frac{\eta}{2}\mathbb{E}\big[\|\nabla f(\mathbf{x}_t)\|^2\big]+\frac{\eta L^2}{2}\mathbb{E}\Big[\Big\|\frac{1}{N}\sum_{i=1}^{N}\mathbf{e}_t^i\Big\|^2\Big]$$

$$\overset{(e)}{\leq} \frac{\eta}{2}\mathbb{E}\big[\|\nabla f(\mathbf{x}_t)\|^2\big]+\frac{\eta^3 L^2}{2}\frac{c}{(1-c)\lambda}(\sigma^2+G^2)$$

where (c) is by Young's inequality; (d) uses Assumption 1 and $\mathbf{z}_t = \mathbf{x}_t - \frac{1}{N}\sum_{i=1}^{N}\mathbf{e}_t^i$; and (e) comes from Lemma 1.

Next, we have

$$\eta\mathbb{E}\big[\langle\nabla f(\mathbf{x}_t),\mathbf{g}_t^i\rangle\big] = \eta\mathbb{E}\Big[\mathbb{E}\big[\langle\nabla f(\mathbf{x}_t),\mathbf{g}_t^i\rangle|\mathbf{x}_t\big]\Big] = \eta\mathbb{E}\big[\|\nabla f(\mathbf{x}_t)\|^2\big]. \qquad (13)$$

To proceed, we have

$$\frac{L}{2}\mathbb{E}\Big[\Big\|\frac{\eta}{N}\sum_{i=1}^{N}\mathbf{g}_t^i+\frac{1}{N}\sum_{i=1}^{N}\zeta_t^i\Big\|^2\Big] \qquad (14)$$

$$\leq L\eta^2\mathbb{E}\Big[\Big\|\frac{1}{N}\sum_{i=1}^{N}\mathbf{g}_t^i-\nabla f(\mathbf{x}_t)+\nabla f(\mathbf{x}_t)\Big\|^2\Big]+L\mathbb{E}\Big[\Big\|\frac{1}{N}\sum_{i=1}^{N}\zeta_t^i\Big\|^2\Big]$$

$$\leq L\eta^2\mathbb{E}\big[\|\nabla f(\mathbf{x}_t)\|^2\big]+L\eta^2\mathbb{E}\Big[\Big\|\frac{1}{N}\sum_{i=1}^{N}\mathbf{g}_t^i-\nabla f(\mathbf{x}_t)\Big\|^2\Big]+L\mathbb{E}\Big[\Big\|\frac{1}{N}\sum_{i=1}^{N}\zeta_t^i\Big\|^2\Big]$$

$$\leq L\eta^2\mathbb{E}\big[\|\nabla f(\mathbf{x}_t)\|^2\big]+\frac{L\eta^2\sigma^2}{N}+L\mathbb{E}\Big[\Big\|\frac{1}{N}\sum_{i=1}^{N}\zeta_t^i\Big\|^2\Big]$$

$$\leq L\eta^2\mathbb{E}\big[\|\nabla f(\mathbf{x}_t)\|^2\big]+\frac{L\eta^2\sigma^2}{N}+\frac{1-c+c/\lambda}{1-c}\frac{2L\delta\theta\eta^2}{N}(\sigma^2+G^2)$$

where the last inequality comes from Lemma 2.

Taking expectation on both sides of (10), and plugging (11), (12), (13), and (14) in, we have

$$\mathbb{E}\big[f(\mathbf{z}_{t+1})-f(\mathbf{z}_t)\big] \leq (-\frac{\eta}{2}+L\eta^2)\mathbb{E}\big[\|\nabla f(\mathbf{x}_t)\|^2\big]$$

$$+\frac{c}{(1-c)\lambda}\frac{L^2\eta^3(\sigma^2+G^2)}{2}+\frac{\eta^2\sigma^2 L}{N}+\frac{1-c+c/\lambda}{1-c}\frac{2\delta\theta\eta^2 L(\sigma^2+G^2)}{N}.$$

Rearranging the terms and dividing both sides with $\eta$, we arrive at

$$(\frac{1}{2}-L\eta)\mathbb{E}\big[\|\nabla f(\mathbf{x}_t)\|^2\big] \leq \frac{\mathbb{E}\big[f(\mathbf{z}_t)-f(\mathbf{z}_{t+1})\big]}{\eta}$$

$$+\frac{c}{(1-c)\lambda}\frac{L^2\eta^2(\sigma^2+G^2)}{2}+\frac{\eta\sigma^2 L}{N}+\frac{1-c+c/\lambda}{1-c}\frac{2\delta\theta\eta L(\sigma^2+G^2)}{N}.$$

Summing over $t$, and using the facts $\mathbf{z}_0=\mathbf{x}_0$ and $f(\mathbf{z}_T)\geq f^*$, we arrive at

$$(\frac{1}{2}-L\eta)\frac{1}{T}\sum_{t=0}^{T-1}\mathbb{E}\big[\|\nabla f(\mathbf{x}_t)\|^2\big] \leq \frac{\big(f(\mathbf{x}_0)-f^*\big)}{\eta T}$$

$$+\frac{c}{(1-c)\lambda}\frac{L^2\eta^2(\sigma^2+G^2)}{2}+\frac{\eta\sigma^2 L}{N}+\frac{1-c+c/\lambda}{1-c}\frac{2\delta\theta\eta L(\sigma^2+G^2)}{N}.$$

Let $\eta = \min\left\{\frac{1}{4L}, \frac{1}{L(\sqrt{\frac{T}{N}}+T^{1/3})}\right\}$. In this case, we have $\frac{1}{2} - L\eta \geq \frac{1}{4}$, $\eta \leq (\frac{1}{LT^{1/3}}$ and $\eta \leq \frac{\sqrt{N}}{L\sqrt{T}}$. Using these three inequalities, we have

$$\frac{1}{4T}\sum_{t=0}^{T-1}\mathbb{E}\big[\|\nabla f(\mathbf{x}_t)\|^2\big] \leq \frac{4L\big(f(\mathbf{x}_0)-f^*\big)}{T} + \frac{L\big(f(\mathbf{x}_0)-f^*\big)}{\sqrt{NT}} + \frac{L\big(f(\mathbf{x}_0)-f^*\big)}{T^{2/3}}$$
$$+ \frac{c}{(1-c)\lambda}\frac{\sigma^2+G^2}{2T^{2/3}} + \frac{\sigma^2}{\sqrt{NT}} + \frac{1-c+c/\lambda}{1-c}\frac{2\delta\theta(\sigma^2+G^2)}{\sqrt{NT}}.$$

If we consider a sufficiently large $T$, i.e, $T > N^3$, we have $\sqrt{NT} < T^{2/3}$. Plugging this inequality in and viewing both $\lambda$ and $c$ as constants, we have

$$\frac{1}{T}\sum_{t=0}^{T-1}\mathbb{E}\big[\|\nabla f(\mathbf{x}_t)\|^2\big] \leq \mathcal{O}\left(\frac{L\big(f(\mathbf{x}_0)-f^*\big)}{\sqrt{NT}}\right) + \mathcal{O}\left(\frac{\sigma^2+G^2}{\sqrt{NT}}\right) + \mathcal{O}\left(\frac{\delta\theta(\sigma^2+G^2)}{\sqrt{NT}}\right).$$

The proof is thus completed. $\qquad\square$

**Lemma 1.** *Suppose that there exist $c \in (0,1)$ and $\lambda > 0$ such that $\delta = \frac{c}{(1+\lambda)(1+\sqrt{\theta})^2}$. It is guaranteed to have*

$$\mathbb{E}\left[\left\|\frac{1}{N}\sum_{i=1}^{N}\mathbf{e}_{t+1}^i\right\|^2\right] \leq \frac{c}{(1-c)\lambda}\eta^2(\sigma^2+G^2), \forall t \geq 0.$$

*Proof.* To start with, we have

$$\mathbb{E}\left[\left\|\frac{1}{N}\sum_{i=1}^{N}\mathbf{e}_{t+1}^i\right\|^2\right] = \frac{1}{N^2}\mathbb{E}\left[\left\|\sum_{i=1}^{N}\mathbf{e}_{t+1}^i\right\|^2\right] \leq \frac{1}{N}\sum_{i=1}^{N}\mathbb{E}\big[\|\mathbf{e}_{t+1}^i\|^2\big]. \tag{15}$$

Conditioning on $\mathbf{g}_t^i$, the randomness only comes from two compressors.

$$\begin{aligned}
\mathbb{E}\big[\|\mathbf{e}_{t+1}^i\|^2\big|\mathbf{g}_t^i\big] &= \mathbb{E}\big[\|\mathbf{p}_t^i - \boldsymbol{\Delta}_t^i + \boldsymbol{\zeta}_t^i\|^2\big|\mathbf{g}_t^i\big] &\tag{16}\\
&\overset{(a)}{\leq} (1+\alpha)\mathbb{E}\big[\|\mathbf{p}_t^i - \boldsymbol{\Delta}_t^i\|^2\big|\mathbf{g}_t^i\big] + \left(1+\frac{1}{\alpha}\right)\mathbb{E}\big[\|\boldsymbol{\zeta}_t^i\|^2\big|\mathbf{g}_t^i\big]\\
&\overset{(b)}{\leq} \left(1+\alpha+\theta+\frac{\theta}{\alpha}\right)\mathbb{E}\big[\|\mathbf{p}_t^i - \boldsymbol{\Delta}_t^i\|^2\big|\mathbf{g}_t^i\big]\\
&\overset{(c)}{\leq} \big(1+\sqrt{\theta}\big)^2\mathbb{E}\big[\|\mathbf{p}_t^i - \boldsymbol{\Delta}_t^i\|^2\big|\mathbf{g}_t^i\big]\\
&\overset{(d)}{\leq} \delta\big(1+\sqrt{\theta}\big)^2\mathbb{E}\big[\|\mathbf{p}_t^i\|^2\big|\mathbf{g}_t^i\big]\\
&= \delta\big(1+\sqrt{\theta}\big)^2\mathbb{E}\big[\|\eta\mathbf{g}_t^i + \mathbf{e}_t^i\|^2\big|\mathbf{g}_t^i\big]\\
&\overset{(e)}{\leq} \left(1+\frac{1}{\lambda}\right)\delta\big(1+\sqrt{\theta}\big)^2\eta^2\|\mathbf{g}_t^i\|^2 + (1+\lambda)\delta\big(1+\sqrt{\theta}\big)^2\mathbb{E}\big[\|\mathbf{e}_t^i\|^2\big|\mathbf{g}_t^i\big]
\end{aligned}$$

where (a) is by Young's inequality, i.e., $\|\mathbf{a}+\mathbf{b}\|^2 \leq (1+\alpha)\|\mathbf{a}\|^2 + (1+\frac{1}{\alpha})\|\mathbf{b}\|^2$ with $\alpha > 0$ to be specified shortly; (b) is the result of (8); (c) is by choosing $\alpha = \sqrt{\theta}$, which minimizes the coefficient; (d) results from Assumption 4; and (e) follows again from Young's inequality.

Then taking expectation w.r.t. $\mathbf{g}_t^i$, we arrive at

$$\mathbb{E}\big[\|\mathbf{e}_{t+1}^i\|^2\big] \leq \underbrace{\left(1+\frac{1}{\lambda}\right)\delta\big(1+\sqrt{\theta}\big)^2}_{:=B_\lambda}\eta^2\mathbb{E}\big[\|\mathbf{g}_t^i\|^2\big] + \underbrace{(1+\lambda)\delta\big(1+\sqrt{\theta}\big)^2}_{:=A_\lambda}\mathbb{E}\big[\|\mathbf{e}_t^i\|^2\big]$$

Summing over $i$, we get

$$\frac{1}{N}\sum_{i=1}^{N}\mathbb{E}\big[\|\mathbf{e}_{t+1}^i\|^2\big] \leq \frac{B_\lambda \eta^2}{N}\sum_{i=1}^{N}\mathbb{E}\big[\|\mathbf{g}_t^i\|^2\big] + \frac{A_\lambda}{N}\sum_{i=1}^{N}\mathbb{E}\big[\|\mathbf{e}_t^i\|^2\big] \tag{17}$$

$$\overset{(f)}{\leq} B_\lambda \eta^2 (\sigma^2 + G^2) + \frac{A_\lambda}{N}\sum_{i=1}^{N}\mathbb{E}\big[\|\mathbf{e}_t^i\|^2\big] \overset{(g)}{\leq} \frac{B_\lambda}{1 - A_\lambda}\eta^2(\sigma^2 + G^2).$$

where (f) follows from Assumption 2; and (g) is obtained by recursively unrolling $\frac{1}{N}\sum_{i=1}^{N}\mathbb{E}\big[\|\mathbf{e}_t^i\|^2\big]$ and using the fact that $\mathbf{e}_0^i = \mathbf{0}$. Given the parameter choice $\delta = \frac{c}{(1+\lambda)(1+\sqrt{\theta})^2}$ for some $c \in (0,1)$, we have $A_\lambda = c$ and $B_\lambda = \frac{c}{\lambda}$. The proof can be completed by using (15). $\qquad\square$

**Lemma 2.** *Choosing the parameters the same as those in Lemma 1, it is guaranteed to have*

$$\mathbb{E}\Big[\big\|\frac{1}{N}\sum_{i=1}^{N}\boldsymbol{\zeta}_t^i\big\|^2\Big] \leq \frac{1 - c + c/\lambda}{1 - c}\frac{2\delta\theta\eta^2(\sigma^2 + G^2)}{N}.$$

*Proof.* Given (7), it is not hard to see that for $i \neq j$,

$$\mathbb{E}\big[\langle \boldsymbol{\zeta}_t^i, \boldsymbol{\zeta}_t^j\rangle|\mathbf{x}_t\big] = 0$$

which implies that

$$\mathbb{E}\Big[\big\|\frac{1}{N}\sum_{i=1}^{N}\boldsymbol{\zeta}_t^i\big\|^2\Big] = \frac{1}{N^2}\sum_{i=1}^{N}\mathbb{E}\big[\|\boldsymbol{\zeta}_t^i\|^2\big]. \tag{18}$$

Then we have from (8) that

$$\mathbb{E}\big[\|\boldsymbol{\zeta}_t^i\|^2|\mathbf{p}_t^i, \boldsymbol{\Delta}_t^i\big] \leq \theta\|\mathbf{p}_t^i - \boldsymbol{\Delta}_t^i\|^2$$

Then taking expectation w.r.t. the randomness of $\mathcal{Q}$, and by Assumption 4, we have

$$\mathbb{E}\big[\|\boldsymbol{\zeta}_t^i\|^2|\mathbf{p}_t^i\big] \leq \delta\theta\|\mathbf{p}_t^i\|^2 = \delta\theta\|\eta\mathbf{g}_t^i + \mathbf{e}_t^i\|^2.$$

Finally, taking expectation w.r.t. $\mathbf{p}_t^i$, we have

$$\mathbb{E}\big[\|\boldsymbol{\zeta}_t^i\|^2\big] \leq \delta\theta\mathbb{E}\big[\|\eta\mathbf{g}_t^i + \mathbf{e}_t^i\|^2\big] \leq 2\delta\theta\eta^2\mathbb{E}\big[\|\mathbf{g}_t^i\|^2\big] + 2\delta\theta\mathbb{E}\big[\|\mathbf{e}_t^i\|^2\big].$$

Summing over $i$, we arrive at

$$\frac{1}{N}\sum_{i=1}^{N}\mathbb{E}\big[\|\boldsymbol{\zeta}_t^i\|^2\big] \leq 2\delta\theta\eta^2\frac{1}{N}\sum_{i=1}^{N}\mathbb{E}\big[\|\mathbf{g}_t^i\|^2\big] + 2\delta\theta\frac{1}{N}\sum_{i=1}^{N}\mathbb{E}\big[\|\mathbf{e}_t^i\|^2\big]$$

$$\leq 2\delta\theta\eta^2(\sigma^2 + G^2) + 2\delta\theta\frac{c}{(1-c)\lambda}\eta^2(\sigma^2 + G^2)$$

$$\leq \frac{1 - c + c/\lambda}{1 - c}2\delta\theta\eta^2(\sigma^2 + G^2)$$

where we use Assumption 2 and (17) in Lemma 1 simultaneously in the second last inequality. The proof is completed by applying (18). $\qquad\square$

## C  PROOF OF THEOREM 2

The proof idea is to define $\mathbf{e}_{t+1}^i := \tilde{\mathbf{e}}_{t+1}^i + \mathbf{q}_{t+1}^i$ and to view $\mathbf{e}_{t+1}^i$ as a single error compressor so that we can reuse the proof of Theorem 1. To see this, define compression error as $\boldsymbol{\zeta}_t^i = \mathbf{e}_{t+1}^i - \mathbf{p}_t^i + \boldsymbol{\Delta}_t^i$. Then we have

$$\mathbb{E}\big[\boldsymbol{\zeta}_t^i|\mathbf{p}_t^i, \boldsymbol{\Delta}_t^i\big] = \mathbb{E}\big[\mathbf{e}_{t+1}^i|\mathbf{p}_t^i, \boldsymbol{\Delta}_t^i\big] - (\mathbf{p}_t^i - \boldsymbol{\Delta}_t^i) \tag{19}$$

$$= \mathbb{E}\big[\tilde{\mathbf{e}}_{t+1}^i|\mathbf{p}_t^i, \boldsymbol{\Delta}_t^i\big] + \mathbb{E}\big[\mathcal{C}(\mathbf{p}_t^i - \boldsymbol{\Delta}_t^i - \tilde{\mathbf{e}}_{t+1}^i)|\mathbf{p}_t^i, \boldsymbol{\Delta}_t^i\big] - (\mathbf{p}_t^i - \boldsymbol{\Delta}_t^i) = \mathbf{0}.$$

where the last equation uses the fact $\mathbb{E}\big[\mathcal{C}(\mathbf{p}_t^i - \boldsymbol{\Delta}_t^i - \tilde{\mathbf{e}}_{t+1}^i)|\mathbf{p}_t^i, \boldsymbol{\Delta}_t^i, \tilde{\mathbf{e}}_{t+1}^i\big] = \mathbf{p}_t^i - \boldsymbol{\Delta}_t^i - \tilde{\mathbf{e}}_{t+1}^i$. This means that $\mathbf{e}_{t+1}^i$ is still an unbiased estimator of $\mathbf{p}_t^i - \boldsymbol{\Delta}_t^i$. However, by adding the compression error back, the variance of $\boldsymbol{\zeta}_t^i$ can be reduced, i.e.,

$$
\begin{aligned}
\mathbb{E}\big[\|\boldsymbol{\zeta}_t^i\|^2|\mathbf{p}_t^i, \boldsymbol{\Delta}_t^i\big] &= \mathbb{E}\big[\|\tilde{\mathbf{e}}_{t+1}^i + \mathcal{C}(\mathbf{p}_t^i - \boldsymbol{\Delta}_t^i - \tilde{\mathbf{e}}_{t+1}^i) - \mathbf{p}_t^i + \boldsymbol{\Delta}_t^i\|^2|\mathbf{p}_t^i, \boldsymbol{\Delta}_t^i\big] \\
&\leq \theta\mathbb{E}\big[\|\tilde{\mathbf{e}}_{t+1}^i - (\mathbf{p}_t^i - \boldsymbol{\Delta}_t^i)\|^2|\mathbf{p}_t^i, \boldsymbol{\Delta}_t^i\big] \\
&= \theta\mathbb{E}\big[\|\mathcal{C}(\mathbf{p}_t^i - \boldsymbol{\Delta}_t^i) - (\mathbf{p}_t^i - \boldsymbol{\Delta}_t^i)\|^2|\mathbf{p}_t^i, \boldsymbol{\Delta}_t^i\big] \\
&\leq \theta^2\|\mathbf{p}_t^i - \boldsymbol{\Delta}_t^i\|^2.
\end{aligned}
\tag{20}
$$

Comparing with (8), one can see that $\theta$ dependence is indeed improved given an accurate $\mathcal{C}$ with $\theta < 1$. Next, we modify Lemmas 1 and 2 to finish the proof.

**Lemma 3.** *Suppose that there exists $c \in (0,1)$ and $\lambda > 0$ such that $\delta = \frac{c}{(1+\lambda)(1+\theta)^2}$. It is guaranteed to have*

$$
\mathbb{E}\Big[\Big\|\frac{1}{N}\sum_{i=1}^N \mathbf{e}_{t+1}^i\Big\|^2\Big] \leq \frac{c}{(1-c)\lambda}\eta^2(\sigma^2 + G^2), \forall t \geq 0.
$$

*Proof.* The proof is basically the same as Lemma 1 except for using (20) rather than (8) in (16) to tighten the bound. Hence, we only highlight the difference here.

$$
\begin{aligned}
\mathbb{E}\big[\|\mathbf{e}_{t+1}^i\|^2\big|\mathbf{g}_t^i\big] &= \mathbb{E}\big[\|\mathbf{p}_t^i - \boldsymbol{\Delta}_t^i + \boldsymbol{\zeta}_t^i\|^2\big|\mathbf{g}_t^i\big] \\
&\leq (1+\alpha)\mathbb{E}\big[\|\mathbf{p}_t^i - \boldsymbol{\Delta}_t^i\|^2|\mathbf{g}_t^i\big] + \Big(1 + \frac{1}{\alpha}\Big)\mathbb{E}\big[\|\boldsymbol{\zeta}_t^i\|^2|\mathbf{g}_t^i\big] \\
&\overset{(a)}{\leq} \Big(1 + \alpha + \theta^2 + \frac{\theta^2}{\alpha}\Big)\mathbb{E}\big[\|\mathbf{p}_t^i - \boldsymbol{\Delta}_t^i\|^2|\mathbf{g}_t^i\big] \\
&\overset{(b)}{\leq} (1+\theta)^2\mathbb{E}\big[\|\mathbf{p}_t^i - \boldsymbol{\Delta}_t^i\|^2|\mathbf{g}_t^i\big] \\
&\leq \delta(1+\theta)^2\mathbb{E}\big[\|\mathbf{p}_t^i\|^2|\mathbf{g}_t^i\big] \\
&= \delta(1+\theta)^2\mathbb{E}\big[\|\eta\mathbf{g}_t^i + \mathbf{e}_t^i\|^2|\mathbf{g}_t^i\big] \\
&\overset{(e)}{\leq} \Big(1 + \frac{1}{\lambda}\Big)\delta(1+\theta)^2\eta^2\|\mathbf{g}_t^i\|^2 + (1+\lambda)\delta(1+\theta)^2\mathbb{E}\big[\|\mathbf{e}_t^i\|^2|\mathbf{g}_t^i\big]
\end{aligned}
\tag{21}
$$

where in (a) we uses (20); (b) chooses $\alpha = \theta$. $\square$

**Lemma 4.** *Choosing parameters the same as those in Lemma 3, it is guaranteed to have*

$$
\mathbb{E}\Big[\Big\|\frac{1}{N}\sum_{i=1}^N \boldsymbol{\zeta}_t^i\Big\|^2\Big] \leq \frac{1 - c + c/\lambda}{1-c}\frac{2\delta\theta^2\eta^2(\sigma^2 + G^2)}{N}.
$$

*Proof.* The proof is exactly the same as Lemma 2 except for using (20) rather than (8). Hence it is omitted here. $\square$

**Proof of Theorem 2.** The rest of proof is the same as Theorem 1, except for i) applying Lemma 3 in (12); and ii) applying Lemma 4 in (14).

## D    PROOF OF THEOREM 3

*Proof.* The basic idea is again to define $\mathbf{e}_{t+1}^i = \tilde{\mathbf{e}}_{t+1}^i + \mathbf{q}_{t+1}^i$ and to view $\mathbf{e}_{t+1}^i$ as a single error compressor so that we can reuse the proof of Theorem 1. Applying (Horváth & Richtárik, 2020, Theorem 3), we have $\mathbf{e}_{t+1}^i$ is unbiased with bounded variance. In particular, we have

$$
\mathbb{E}\big[\mathbf{e}_{t+1}^i - (\mathbf{p}_t^i - \boldsymbol{\Delta}_t^i)|\mathbf{p}_t^i, \boldsymbol{\Delta}_t^i\big] = \mathbb{E}\big[(\tilde{\mathbf{e}}_{t+1}^i + \mathbf{q}_{t+1}^i) - (\mathbf{p}_t^i - \boldsymbol{\Delta}_t^i)|\mathbf{p}_t^i, \boldsymbol{\Delta}_t^i\big] = \mathbf{0}
\tag{22}
$$

$$
\mathbb{E}\big[\|\mathbf{e}_{t+1}^i - (\mathbf{p}_t^i - \boldsymbol{\Delta}_t^i)\|^2|\mathbf{p}_t^i, \boldsymbol{\Delta}_t^i\big] = \mathbb{E}\big[\|(\tilde{\mathbf{e}}_{t+1}^i + \mathbf{q}_{t+1}^i) - (\mathbf{p}_t^i - \boldsymbol{\Delta}_t^i)\|^2|\mathbf{p}_t^i, \boldsymbol{\Delta}_t^i\big]
\tag{23}
$$
$$
\leq \delta\theta\|\mathbf{p}_t^i - \boldsymbol{\Delta}_t^i\|^2.
$$

Therefore, we can reuse the derivation in Theorem 1. The only change we need is to modify Lemmas 1 and 2 to Lemmas 5 and 6, respectively. After applying these modified lemmas, Theorem 3 directly follows.

$\square$

**Lemma 5.** *Suppose that there exists $c \in (0,1)$ and $\lambda > 0$ such that $\delta = \frac{c}{(1+\lambda)(1+\sqrt{\theta})^2}$. It is guaranteed to have*

$$\mathbb{E}\Big[\Big\|\frac{1}{N}\sum_{i=1}^{N}\mathbf{e}_{t+1}^i\Big\|^2\Big] \leq \frac{c}{(1-c)\lambda}\eta^2(\sigma^2+G^2), \forall t \geq 0.$$

*Proof.* The proof is basically the same as Lemma 1, hence we only highlight the different steps in (16) here. Define $\boldsymbol{\zeta}_t^i = \mathbf{e}_{t+1}^i - \mathbf{p}_t^i + \boldsymbol{\Delta}_t^i$, we have

$$\mathbb{E}\big[\|\mathbf{e}_{t+1}^i\|^2\big|\mathbf{g}_t^i\big] = \mathbb{E}\big[\|\mathbf{p}_t^i - \boldsymbol{\Delta}_t^i + \boldsymbol{\zeta}_t^i\|^2\big|\mathbf{g}_t^i\big] \qquad (24)$$

$$\leq (1+\alpha)\mathbb{E}\big[\|\mathbf{p}_t^i - \boldsymbol{\Delta}_t^i\|^2\big|\mathbf{g}_t^i\big] + \Big(1+\frac{1}{\alpha}\Big)\mathbb{E}\big[\|\boldsymbol{\zeta}_t^i\|^2\big|\mathbf{g}_t^i\big]$$

$$\overset{(a)}{\leq} \Big(1+\alpha+\delta\theta+\frac{\delta\theta}{\alpha}\Big)\mathbb{E}\big[\|\mathbf{p}_t^i - \boldsymbol{\Delta}_t^i\|^2\big|\mathbf{g}_t^i\big]$$

$$\overset{(b)}{\leq} \big(1+\sqrt{\delta\theta}\big)^2\mathbb{E}\big[\|\mathbf{p}_t^i - \boldsymbol{\Delta}_t^i\|^2\big|\mathbf{g}_t^i\big]$$

$$\overset{(c)}{\leq} \big(1+\sqrt{\theta}\big)^2\mathbb{E}\big[\|\mathbf{p}_t^i - \boldsymbol{\Delta}_t^i\|^2\big|\mathbf{g}_t^i\big]$$

where (a) is the result of (23); (b) is by choosing $\alpha = \sqrt{\delta\theta}$; and (c) follows from $\delta < 1$. $\square$

**Lemma 6.** *Choosing the parameters the same as those in Lemma 5, it is guaranteed in iConEF-v2 to have*

$$\mathbb{E}\Big[\Big\|\frac{1}{N}\sum_{i=1}^{N}\boldsymbol{\zeta}_t^i\Big\|^2\Big] \leq \frac{1-c+c/\lambda}{1-c}\frac{2\delta^2\theta\eta^2(\sigma^2+G^2)}{N}.$$

*Proof.* The proof is almost the same as that of Lemma 2, hence we only highlight the differences.

From (23), we have

$$\mathbb{E}\big[\|\boldsymbol{\zeta}_t^i\|^2\big|\mathbf{p}_t^i\big] \leq \delta^2\theta\|\mathbf{p}_t^i\|^2 = \delta^2\theta\|\eta\mathbf{g}_t^i + \mathbf{e}_t^i\|^2.$$

Finally, taking expectation w.r.t. $\mathbf{p}_t$, we have

$$\mathbb{E}\big[\|\boldsymbol{\zeta}_t^i\|^2\big] \leq \delta^2\theta\mathbb{E}\big[\|\eta\mathbf{g}_t^i + \mathbf{e}_t^i\|^2\big] \leq 2\delta^2\theta\eta^2\mathbb{E}\big[\|\mathbf{g}_t^i\|^2\big] + 2\delta^2\theta\mathbb{E}\big[\|\mathbf{e}_t^i\|^2\big]$$

$$\leq \frac{1-c+c/\lambda}{1-c}2\delta^2\theta\eta^2(\sigma^2+G^2).$$

where we use Assumption 2 and Lemma 5 simultaneously in the last inequality With this modification, Lemma 6 follows directly. $\square$

## E  PROOF OF THEOREM 4

We focus on ConEF. The proofs for iConEF-v1 and iConEF-v2 follow the same steps (simply by defining $\mathbf{e}_{t+1}^i = \tilde{\mathbf{e}}_{t+1}^i + \mathbf{q}_{t+1}^i$) and hence are omitted here. Recall that in (9) we have $\mathbf{z}_{t+1} = \mathbf{z}_t - \frac{\eta}{N}\sum_{i=1}^{N}\mathbf{g}_t^i - \frac{1}{N}\sum_{i=1}^{N}\boldsymbol{\zeta}_t^i$ with $\mathbf{z}_t = \mathbf{x}_t - \frac{1}{N}\sum_{i=1}^{N}\mathbf{e}_t^i$. This implies that

$$\mathbf{x}_{t+1} - \frac{1}{N}\sum_{i=1}^{N}\mathbf{e}_{t+1}^i = \mathbf{x}_t - \frac{1}{N}\sum_{i=1}^{N}\mathbf{e}_t^i - \frac{\eta}{N}\sum_{i=1}^{N}\mathbf{g}_t^i - \frac{1}{N}\sum_{i=1}^{N}\boldsymbol{\zeta}_t^i.$$

Conditioned on $\{\mathbf{x}_t, \{\mathbf{e}_t^i\}_i, \{\mathbf{g}_t^i\}_i\} := \mathcal{P}_t$ and only take expectation w.r.t. the randomness of $\mathcal{C}$, we have

$$\mathbb{E}\Big[\mathbf{x}_{t+1} - \frac{1}{N}\sum_{i=1}^{N}\mathbf{e}_{t+1}^i | \mathcal{P}_t\Big] = \mathbb{E}\Big[\mathbf{x}_t - \frac{1}{N}\sum_{i=1}^{N}\mathbf{e}_t^i - \frac{\eta}{N}\sum_{i=1}^{N}\mathbf{g}_t^i \Big| \mathcal{P}_t\Big] - \mathbb{E}\Big[\frac{1}{N}\sum_{i=1}^{N}\boldsymbol{\zeta}_t^i \Big| \mathcal{P}_t\Big]$$

$$= \mathbb{E}\Big[\mathbf{x}_t - \frac{1}{N}\sum_{i=1}^{N}\mathbf{e}_t^i - \frac{\eta}{N}\sum_{i=1}^{N}\mathbf{g}_t^i \Big| \mathcal{P}_t\Big].$$

Then we further take expectation w.r.t. $\mathcal{P}_t$ and unroll $\mathbb{E}\big[\mathbf{x}_t - \frac{1}{N}\sum_{i=1}^{N}\mathbf{e}_t^i\big]$, we can get

$$\mathbb{E}\Big[\mathbf{x}_{t+1} - \frac{1}{N}\sum_{i=1}^{N}\mathbf{e}_{t+1}^i\Big] = \mathbf{x}_0 - \eta\mathbb{E}\Big[\sum_{\tau=0}^{t}\frac{1}{N}\sum_{i=1}^{N}\mathbf{g}_\tau^i\Big].$$

Given that $\mathbf{x}_0 \in \mathcal{R}(\mathbf{A}^\top)$, and $\mathbf{g}_t^i \in \mathcal{R}(\mathbf{A}^\top), \forall t$, it is straightforward to see that $\mathbb{E}\big[\mathbf{x}_{t+1} - \frac{1}{N}\sum_{i=1}^{N}\mathbf{e}_{t+1}^i\big] \in \mathcal{R}(\mathbf{A}^\top)$. Hence the proof is completed.

## F  PARTIAL CONEF/EFSGD

To utilize memory in the most efficient manner when adopting count sketches as the error compressor, we introduce a variant ConEF, termed partial ConEF in Alg. 4. Partial ConEF is adapted from partial EFSGD (Abdi & Fekri, 2020), which can be recovered when $\mathcal{C}$ in line 9 is an identity mapping. The main difference with ConEF is the introduction of the scaler $\beta \in [0, 1)$, highlighted in blue. Partial EFSGD/ConEF adds *part* of the compression error, i.e., $(1 - \beta)\mathbf{e}_t^i$, to the stochastic gradient before compressing; see line 5. The remained local error $\beta\mathbf{e}_t^i$ is kept and included in the update of $\mathbf{e}_{t+1}^i$ in line 9 for the use of next iteration. The convergence of partial ConEF is straight forward when combining our proof with (Abdi & Fekri, 2020).

---

**Algorithm 4** Partial EFSGD / partial ConEF

---
1: **Initialize:** $\mathbf{x}_0 \in \mathbb{R}^d, \mathbf{e}_0^i = \mathbf{0} \in \mathbb{R}^d, \forall i, \eta, \beta \in [0, 1)$
2: **for** $t = 0, 1, \ldots, T - 1$ **do**
3:     assert $\mathbf{x}_t = \mathbf{x}_t^i$ for every worker $i$
4:     **for** worker $i = 1, \ldots N$ in parallel **do**
5:         $\mathbf{p}_t^i = \eta\mathbf{g}_t^i + (1 - \beta)\mathbf{e}_t^i$
6:         $\boldsymbol{\Delta}_t^i = \mathcal{Q}(\mathbf{p}_t^i)$
7:         $\boldsymbol{\Delta}_t = \text{Aggregate}(\boldsymbol{\Delta}_t^i, \forall i)$
8:         $\mathbf{x}_{t+1} = \mathbf{x}_t - \boldsymbol{\Delta}_t$
9:         $\mathbf{e}_{t+1}^i = \mathcal{C}(\beta\mathbf{e}_t^i + \mathbf{p}_t^i - \boldsymbol{\Delta}_t^i)$
10:     **end for**
11: **end for**

---

The partial ConEF update in line 9 is helpful when $\mathcal{C}$ is count sketch. In particular, we can rewrite line 9 as $\mathbf{e}_{t+1}^i = \mathcal{C}(\mathbf{e}_t^i - (1 - \beta)\mathbf{e}_t^i + \mathbf{p}_t^i - \boldsymbol{\Delta}_t^i)$. This means that the compression error in $\mathbf{e}_t^i$ is also scaled down or subtracted. This is helpful for suppressing the accumulation of compression error.

Partial ConEF benefits from the linearity of count sketch since the decompression of $\mathbf{e}_t^i$ can be avoided when updating $\mathbf{e}_{t+1}^i$ in line 9. One can simply compress $\mathbf{p}_t^i - \boldsymbol{\Delta}_t^i$ and add it to the scaled $\mathbf{e}_t^i$. Moreover, auxiliary variables in global memory can be avoided by taking advantage of the shared memory of the GPU. Compared to a nonlinear error compressor $\mathcal{C}$, count sketch further reduces the burden on runtime.

## G  MORE ON NUMERICAL EXPERIMENTS

Table 2: $\beta$ vs. sketch size for convergence for unscaled random block $k$ gradient compressor

| $\beta$ | 0.1 | 0.2 | 0.3 | 0.4 | 0.5 | 0.6 | 0.7 | 0.8 | 0.9 |
|---|---|---|---|---|---|---|---|---|---|
| smallest sketch size | − | 0.9 | 0.9 | 0.8 | 0.7 | 0.5 | 0.3 | 0.3 | 0.1 |

Experiments are repeated 3 times and averaged results are reported. In the implementation of count sketches, the tensor trick can be employed to avoid computing too many hashes, which improves the runtime. In the tensor trick, instead of using element-wise compressing, we view the error vector as a matrix, and treat one row or column as an "entry" of the compressed vector, resulting in a tensor sketch (matrix when $v = 1$). More specifically, suppose the error vector lives in $\mathbb{R}^d$, whose matrix view is $\mathbb{R}^{n \times m}$ with $d = mn$. It takes $n$ (resp. $m$) hash computation in a row- (resp. column-) tensor trick, while $mn$ hashes are needed without this trick. Differences on test accuracy between row- or column-based tensor tricks are not observed; see Fig. 7.

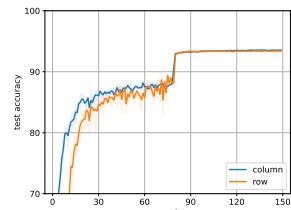

Figure 7: ConEF with row or column hashes has similar performance.

### G.1 PARAMETER CHOICES FOR RQ1

Experiments in this subsection is carried out using a single Amazon EC2 p3.8xlarge instance with 4 GPUs. The goal here is to simulate a parameter-server setting since we do not have a real testbed.

**ResNet18 on CIFAR10.** The model is trained for 150 epochs with weight decay $10^{-4}$ and momentum 0.9. We decrease the learning rate by a factor of 10 at epochs 80 and 120. Overall batchsize 128 (32 per worker). And the initial learning rate for SGD is tuned from $\{0.1, 0.075, 0.05, 0.025\}$, with final choice of 0.1. EFSGD, ConEF, and QSGD use the same learning rate. In ConEF-CS we use $\beta = 0.9$.

**MobileNetV2 on CIFAR10.** The model is trained for 150 epochs with weight decay $4 \times 10^{-5}$ and momentum 0.9. We decrease the learning rate by a factor of 10 at epochs 80 and 120. Overall batchsize is 128 (32 per worker). And the initial learning rate for SGD is tuned from $\{0.1, 0.075, 0.05, 0.025\}$, with final choice of 0.1. The same learning rate is used for EFSGD and ConEF. $\beta = 0.9$ is adopted in ConEF-CS.

**Choices of $\beta$ for allreducable gradient compressors.** While the parameter guidances for scaled-sign gradient compressor are discussed, we also include the choice of $\beta$ and the smallest sketch size for convergence in Table. 2 when the gradient compressor $\mathcal{Q}$ is unscaled random-block-$k$, which is allreducable. We choose $k$ such that 90% of communication overhead is reduced. Similar to scaled-sign, it is observed that a larger $\beta$ leads to more saved memory. Hence, we recommend to choose $\beta = 0.9$ or even larger.

### G.2 PARAMETER CHOICES FOR RQ2

The experiments are carried out using 4 Amazon EC2 p3.8xlarge instances (16 GPUs in total).

**ResNet18 on CIFAR10.** The model is trained for 150 epochs with weight decay $10^{-4}$ and momentum 0.9. We decrease the learning rate by a factor of 10 at epochs 80 and 120. We focus on small batchsize setting with 16 data sample per worker (256 in total). The initial learning rate for SGD is tuned from $\{0.1, 0.075, 0.05, 0.025\}$, finally chosen as 0.1. EFSGD, ConEF, and hEFSGD use the same learning rate. ConEF-CS with 60% memory saving uses $\beta = 0.85$; while its 90% memory saving relies on $\beta = 0.9$.

### G.3 PARAMETER CHOICES FOR RQ3

All experiments are carried out using 4 Amazon EC2 p3.8x large instances (16 GPUs in total).

**ResNet18 on CIFAR10.** We train this model for 120 epochs and decrease the learning rate by 10 at epochs 60 and 100. Momentum is chosen as 0.9 and weight decay is set to $10^{-4}$. To save memory, we focus on small batch setting with batchsize 16 per GPU (hence 256 in total). The learning rate for SGD is tuned from $\{0.05, 0.1, 0.2\}$, and 0.2 is chosen. We use the same learning rate in EFSGD

Table 3: Numerical results for ResNet18

| Algorithm | test accuracy | runtime (min) | memory saving (MB) |
|---|---|---|---|
| SGD | 93.20 | 54.83 | - |
| rbSGD | 86.58 | 40.12 | 48.2 |
| EFSGD | 92.52 | 40.41 | 0 |
| ConEF 60% | 92.53 | 42.45 | 31.5 |
| ConEF 90% | 92.02 | 42.45 | 44.1 |

Table 4: Numerical results for WRN-28-10

| Algorithm | test accuracy | runtime (min) | memory saving (MB) |
|---|---|---|---|
| SGD | 96.12 | 139.70 | - |
| rbSGD | 82.92 | 70.88 | 122.1 |
| EFSGD | 95.28 | 71.13 | 0 |
| ConEF 60% | 96.00 | 72.11 | 76.2 |
| ConEF 90% | 95.23 | 71.98 | 101.5 |

and ConEF-CS. For ConEF-CS with $60\%$ saved memory, we choose $\beta = 0.9$; and for ConEF-CS with $90\%$ memory saving we use $\beta = 0.95$. For rbSGD, we test step sizes from $\{0.05.0.1, 0.2\}$ and $0.1$ is chosen.

**WideResNet28-10 on CIFAR10.** The model is trained 150 epochs. Learning rate is decreased by 5 on epochs 50, 90, and 120. We set weight decay as $5 \times 10^{-4}$ and momentum 0.9. Nesterov momentum is also adopted. A small batchsize 16 per GPU (256 in total) is considered for memory saving. The learning rate for SGD is tuned from $\{0.05, 0.1, 0.2\}$, where $0.1$ performs the best. EFSGD and ConEF-CS adopt the same learning rate. For ConEF-CS with $60\%$ saved memory, we choose $\beta = 0.75$; and for ConEF-CS with $90\%$ memory saving we use $\beta = 0.995$. For rbSGD, we test step sizes from $\{0.001, 0.005, 0.05.0.1\}$ and $0.005$ is the final choice.

**LSTM on Wikitext-2.** We use a 2-layer LSTM with 672 hidden units and 672-dimension word embeddings. We set BPTT as 35, and clip the gradient norm to 0.25. The momentum is chosen as 0.9, aided with Nesterov momentum as well. Dropout rate is chosen as 0.5. We also consider a small batchsize setting, e.g., 2 per GPU (32 in total). The model is trained for 60 epochs where the learning rate is decreased by 4 at epochs $15, 30$, and $45$. The initial learning rate for SGD is tuned from $\{0.5, 1, 2, 4\}$, and $2$ is chosen. The same learning rate schedule is adopted for EFSGD and ConEF-CS. For ConEF-CS with $60\%$ saved memory, we choose $\beta = 0.8$; and for ConEF-CS with $80\%$ memory saving we use $\beta = 0.9$. For rbSGD, we test step sizes from $\{0.5, 1, 2, 4\}$ and $1$ is chosen.

### G.4 ADDITIONAL EXPERIMENTS

**Adam type algorithms on a transformer.** A single layer transformer for English-Germen machine translation is trained on Multi30k dataset (Elliott et al., 2016). In particular, we use a 512 dimensional embedding layer. Both encoder and decoder have 3 layers with hidden dimension 512 per layer and 8-head attention is adopted. The initial learning rate for ADAM and other algorithms is set as $10^{-4}$. The transformer is trained for 30 epochs, where the learning rate is decreased by 4 at the end of epochs 15 and 25. Batchsize is set to 128 and dropout ratio is set as 0.1. The biased gradient compressor is chosen as unscaled random block $k$ to reduce $90\%$ communication overhead.

Adam with an allreducable and unbiased gradient compressor (rb-Adam) struggles in terms of training loss. EF-Adam converges even slower than its ConEF counterpart. This once again demonstrates

Table 5: Numerical results for LSTM

| Algorithm | perplexity | runtime (hour) | memory saving (MB) |
|---|---|---|---|
| SGD | 92.72 | 7.57 | - |
| rbSGD | 120.07 | 1.45 | 262.5 |
| EFSGD | 98.29 | 1.49 | 0 |
| ConEF 60% | 97.76 | 1.51 | 180.3 |
| ConEF 80% | 98.59 | 1.50 | 220.2 |

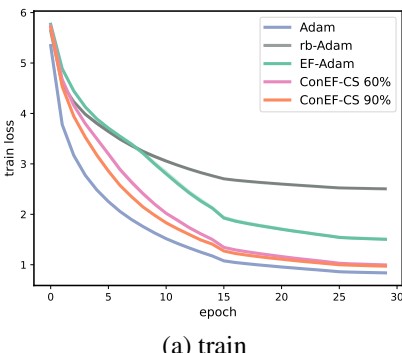 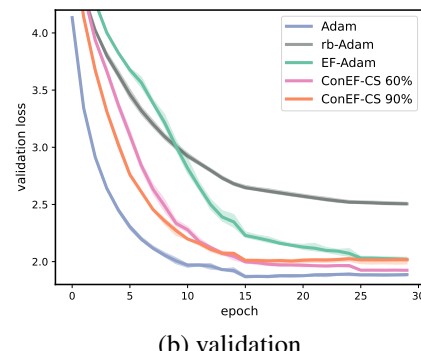

(a) train

(b) validation

Figure 8: Performance of ConEF with unscaled random block gradient compressor on a transformer.

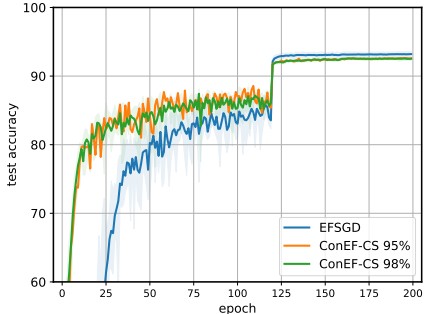

Figure 9: More aggressively saved memory with random-$k$ gradient compressor.

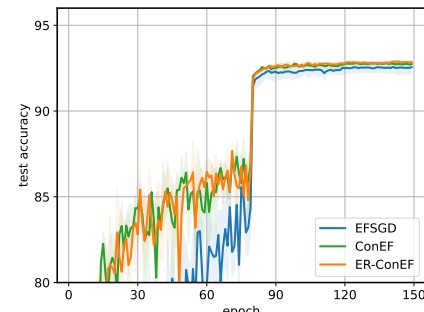

Figure 10: Test of ConEF-CS with error reset using random-$k$ gradient compressor.

that EF-Adam does not take full advantage of memory. ConEF with $60\%$ and $90\%$ memory saved obtain similar training loss both slightly larger than Adam, but there is a gap on validation.

Regarding validation loss, rb-Adam fails to match with other tested algorithms. With $90\%$ memory saved, ConEF-CS still outperforms EF-Adam slightly. ConEF-CS with $60\%$ saved memory has a comparable performance to vanilla Adam, and both are much better than EF-Adam. These observations suggest that ConEF is an improved alternative of error feedback to endow Adam type optimizers with communication and memory efficiency in distributed training. Lastly, the validation gap between ConEF with $60\%$ and $90\%$ memory saved intuitively makes sense, since as shown in Theorem 4, the best generalization error can be achieved when subtracting averaged local error vectors from current model. The count sketch in ConEF-CS introduces larger compression error when more memory is saved (small sketch size), thus losing more on its generalization behavior.

**Random-$k$ gradient compressor with more memory saving.** We also test the unscaled random-$k$ gradient compressor on ResNet-18 with CIFAR10, where $90\%$ communication burden is reduced. As shown in Fig. 9, EFSGD has a test accuracy of $93.42$, while ConEF-CS with $w = 0.05$ ($95\%$ memory saving) and $w = 0.02$ ($98\%$ memory saving) achieve a test accuracy of $92.70$ and $92.68$. Hence, it is possible to have an aggressive memory saving with less than $1$ loss on test accuracy. In addition, it is observed that the compressed error vector is helpful to get a faster convergence initially. But after decreasing the step size, it has negative impact resulting in the accuracy drop.

**iConEF with a random-$k$ gradient compressor.** To validate the improvement of iConEF over ConEF, we train a ResNet-18 on CIFAR10, and the results are reported in Fig. 6. A random-$k$ gradient compressor reducing $90\%$ communication overhead is adopted. For ConEF, we use a $s = 64$-level stochastic quantizer as the error compressor. iConEF-v2 is considered in this case, and

we use top-$k$ compressor (5% elements) as the first error compressor, and $s = 64$-level stochastic quantizer as the second error compressor. As shown in Fig. 6, although ConEF has a similar performance as EFSGD, iConEF-v2 outperforms both of them, validating its efficiency. iConEF-v1 is also included in Fig. 6. In this case, both error compressors are chosen as $s = 256$-level quantizer. iConEF-v1 converges much faster comparing with EFSGD initially, and it has a similar test accuracy after learning rate is decreased.

**ConEF-CS can be used easily with error reset (ER).** As discussed in Section 4.2.2, error reset (Basu et al., 2019; Xie et al., 2020) can be applied directly when the error compressor $\mathcal{C}$ is count sketch. Thanks to the fact that count sketch is allreducable and small in size, ER can be done through allreducing the compressed error among all workers every a few iterations, which in our case is 512. ER slightly improves the test accuracy of ConEF-CS by 0.1 as observed in Fig. 10. More specifically, we train a ResNet-18 on CIFAR10 with unscaled random-$k$ gradient compressor to reduce 90% communication overhead, and choose $\beta = 0.9$ with a count sketch that saves 80% memory. As shown here, both ConEF and ER-ConEF outperform EFSGD, and ER-ConEF has the best test accuracy.

