# OpenReview forum: "Contractive error feedback for gradient compression"
_ICLR.cc/2022/Conference — ICLR 2022 Submitted_

### Official Review · Reviewer_2Hku · 2021-11-01

**Correctness:** 3
**Technical Novelty And Significance:** 2
**Empirical Novelty And Significance:** 3
**Recommendation:** 5
**Confidence:** 4

**Main Review:**

Regarding the method/novelty

1. The author propose to a simple technique to compress the local error either once or twice. It is interesting to see that compressing twice improves the convergence from $\mathcal(O)(\theta)$ to $\mathcal(O)(\theta^2)$ by doubling the memory cost to store the local error. However, the extra computation overheads are not compared. The author did not show the tradeoff between actual training speed v.s. extra memory cost.

2. Closely related works are not compared such as compression of the error [1] and faster training [2].

Regarding the experiment
1. The most important issue is about the unrealistic settings of the experiment.

1.1 4 GPUs is far from convincing to demonstrate the scalability of the proposed method and a fair comparison with other baselines. Standard settings would be 16 GPUs with larger-scale datasets like ImageNet.

1.2 The batch size settings unnecessarily enlarge the communication/computation ratio. A larger local batch size (e.g. b=128 with learning rate warmup) on each worker should be set for training small models like ResNet-18 on CIFAR-10.

1.3 The author mentions 1) growing model size and 2) federated learning as motivation. However, All models considered in experiments are very small. There are no federated learning experiments nor corresponding theoretical analysis.

2. Is compressing the local error really necessary? The proposed methods decreases the performance both theoretically and empirically. There is obvious performance decrease most figures for the proposed method. However, what is the real run-time memory reduction? The model size of ResNet-18 on CIFAR-10 should be trivial compared with the activations.

A suggestion would be summarizing the compression ratio of the gradient and the local error, overall memory reduction (instead of the reduction of local error), and final model performance in a table to clearly show the benefits/trade-off of the proposed method.

References

[1] Xie, C., Zheng, S., Koyejo, O.O., Gupta, I., Li, M. and Lin, H., 2020. Cser: Communication-efficient sgd with error reset. Advances in Neural Information Processing Systems, 33.

[2] Xu, A., Huo, Z. and Huang, H., 2021, January. Step-Ahead Error Feedback for Distributed Training with Compressed Gradient. In Proceedings of the AAAI Conference on Artificial Intelligence (Vol. 35, No. 12, pp. 10478-10486).

**Summary Of The Paper:**

The authors propose to compress the local error in communication-efficient distributed training to reduce the memory to store the local error.

The paper also proposes to compress the local error twice, which doubles the memory cost to store the local error but improves the convergence compared with compressing once.

Experiments show the training performance of the proposed method with different compression ratio. But the settings are unrealistic and can not validate the claims.

**Summary Of The Review:**

The need to compressing the local error in distributed training is not well justified in the experiments. The proposed application scenarios of training large models and federated learning are not demonstrated by the experiments. Besides, the scalability regarding the number of workers and the model size is not very convincing.

---

### Official Review · Reviewer_HC53 · 2021-11-02

**Correctness:** 3
**Technical Novelty And Significance:** 2
**Empirical Novelty And Significance:** 2
**Recommendation:** 3
**Confidence:** 4

**Main Review:**

I see two significant issues with this work.
1. It has been previously mentioned by Ramesh et al. ("Zero-shot text-to-image generation", 2021) that error feedback can be optimized if we accumulate the gradients directly into the error buffer. For instance, in PyTorch one manually sets the gradients to 0 after each gradient computation by calling .zero_grad() but instead we could keep the error vector in this memory space, and then the gradient is added directly there, and no extra memory is used. This small observation makes me wonder why the tools from this work are required at all.
2. Even if we ignore the previous issue, the promised efficiency is very low. Let us say that we take a compression with theta=10. For rand-k or top-k it means that we keep about 10% of coordinates, which is indeed an improvement. However, this may imply a 10 times slower convergence. This is a very big price to pay for a somewhat small reduction in memory use. Since compression operators, such as PowerSGD, usually lead to a speed-up that is smaller than 10x, it may mean that it's better for us to not use compression at all.

I found the iConEF algorithms an interesting addition to the paper story but the theoretical improvement seems to hold only for very niche compressors. The improvement is significant only when the compression ratio is close to 0. This corresponds to almost no compression, and requires to compute an extra compression operator, which make these algorithms unattractive.

The experiments do not seem to be done in the considered setting of large models that require huge memory, but they seem fine to me as a proof of concept. I believe this work should be mainly judged based on its theoretical results.

--Minor--
1. The notation for the objective is inconsistent. In problem (1), the authors write f(x; \xi), but in the paragraph right after that they write f(x, \xi). Please use either a comma or a semicolon throughout the work.
2. The comment "Biased gradient compressors, on the other hand, have been more successful in practice" seems like a stretch to me. As far as I know, the efficiency of PowerSGD is more due to its support of Allreduce.
3. It is also not true that "Biased gradient compressors <...> support a more impressive compression ratio": For instance, top-k and random-k have exactly the same compression ratio, simiarly to how top-k eigenvalues gives the same compression ratio as random-k eigenvalues.
4. "which is typically contractive thus more robust to the gradient variance in a
small batch size setting." I do not see how "contactive" implies "more robust to the gradient variance". The same claim is stated again a bit later and, as far as I can see, without any justification. Please add some supporting reference/evidence or remove this claim.
5. The "Aggregate" operation in the formulation of the algorithms is pretty vague. Even though it is clarified in the text, I'd find it more natural to write this operation as the standard average directly in Algorithms 1 and 2.
6. The training of Resnet18 on Cifar10 is a bit suboptimal. One can achieve test accuracy above 95% by training for 200 or 300 epochs with cosine annealing learning rate schedule.

--Typos--
"state of the art method" => "state-of-the-art method"
"EF based methods" => "EF-based methods"


**Summary Of The Paper:**

Motivated by the memory limitations of large-scale training, this paper proposes a new modification to the error-feedback algorithm for parallel optimization. In particular, the authors suggest compressing the error vector with a separate compressor, which results in the ConEF algorithm. Under the assumption that this compressor is unbiased and has compression ratio theta, the authors prove that the convergence rate has some terms of the same order as in error feedback and others get multiplied by a factor of theta.
In addition, the authors propose the iConEF algorithms that introduce an additional error feedback for the extra compressor. This leads to yet another sequence of compressed vectors that need to be maintained. On the other hand, if theta<1 (or delta<1), the rate will become theta (or delta) times better.

**Summary Of The Review:**


The results in this work are rigorous and the motivation is well explained. The theory is done under reasonable assumptions and the rates are close to what is expected. Unfortunately, there are a few big issues with the proposed solution. First of all, there seems to be a way to make EF memory-efficient without any algorithmic changes. Secondly, the guarantees for the method imply that the methods are significantly slower than standard EF and may be even slower in runtime than SGD without compression. Finally, the experiments may not be the best possible, but I personally think that this is not a big issue as the main contribution is to propose new methods and analyze their convergence.

---

### Official Review · Reviewer_VhWp · 2021-11-03

**Correctness:** 2
**Technical Novelty And Significance:** 2
**Empirical Novelty And Significance:** 2
**Recommendation:** 3
**Confidence:** 3

**Main Review:**

Strengths:
- I have not seen before that a multi-level error correction procedure is applied. This seems to be novel and interesting.

Weaknesses:
- I did not understand most theorems, but it is probably because I am not familiar with the theoretical background. Other reviewers probably have more to say about this.
- The paper is difficult to read. It feels like it is lacking structure between sections. Often I found myself reading a section and wondering how did I end up here and why. I think this can be fixed relatively easily by rewriting and restructuring the paper to guide the reader and explain what is coming next and why.
- The model contains several statements which I think are untrue or not specific enough to be true in most contexts. In particular, memory problems due to gradients are usually not a problem on GPUs due to ZeRO-3 (fully shared parallelism). ZeRO-3 allows the training of models with 10bn parameters or more on relatively few GPUs without a large communication overhead. While the approach in this paper is still valuable, it should be set into context why it is significant
- The presentation of the results is not easy to inspect. I would rather have tables with final performance and speedup/runtime numbers. The charts are too small to really see the differences in methods
- I do not understand the motivation for the sketch count algorithm. Expanding on this would make it clearer
- All the experiments are done on CIFAR-10 and the models have relatively few parameters (small gradients). I do not think this is the best dataset to showcase this method. The current results feel too thin to make accurate judgments about this method and how it fits into the current and future literature

===============================UPDATE================================

Some of my concerns have been alleviated in particular the transformer results in the appendix (that I overlooked) are convincing. I think this paper still has issues that need to be resolved. I think the work is better positioned as communication efficient training rather than training to save memory. There are indeed use-cases where this approach could outperform current approaches that are communication efficient and save memory such as ZeRO.

To me, it feels the rebuttal has been fruitful and could lead to a much better paper. I think it is better that the authors take their time to improve the paper and resubmit it to the next conference. As such, I will keep my current score.

Due to lacking expertise, I am not able to judge the theoretical part of this paper and I would refer to other reviewers for that part of the paper.

**Summary Of The Paper:**

The paper introduces a two-stage error correction procedure for approximated (doubly compressed) gradients. Multiple theorems are derived to prove convergence and scaling properties. The algorithm can be combined with any compression algorithm for the gradient. The algorithm is applied to CIFAR-10 with results showing favorable results.


**Summary Of The Review:**

Overall, this paper is difficult to read and understand, the experiments on CIFAR-10 are not convincing enough, and some claims are underspecified. The innovation of multi-level error correction does not outway these negative factors and as such I recommend rejection.

---

### Decision · Program_Chairs · 2022-01-20

**Decision:**

Reject

**Comment:**

The reviewers initially struggled to position this contribution in terms of usefulness. During the discussion phase, it became (more) clear that the proposed method is best used to reduce the communication overhead of ZeRO3. While the integration of this work and ZeRO hasn't been attempted yet, the authors claim that this work "clears the theoretical barrier". From that point of view, the reviewers were not satisfied with the guarantees of the method, arguing that the resulting algorithm is slower than standard EF and could suffer in terms of runtime (when one factors the cost of compression) even when compared to standard uncompressed SGD. Overall, the discussion greatly improved the paper, although directly integrating ConEF with ZeRO could be even more convincing.